# Protein Design with Guided Discrete Diffusion

Nate Gruver[*1]   Samuel Stanton[*2]   Nathan Frey[2]   Tim G. J. Rudner[1]   Isidro Hotzel[3]

Julien Lafrance-Vanasse[3]   Arvind Rajpal[3]   Kyunghyun Cho[1,2]   Andrew Gordon Wilson[1]

## Abstract

A popular approach to protein design is to combine a generative model with a discriminative model for conditional sampling. The generative model samples plausible sequences while the discriminative model guides a search for sequences with high fitness. Given its broad success in conditional sampling, classifier-guided diffusion modeling is a promising foundation for protein design, leading many to develop guided diffusion models for structure with inverse folding to recover sequences. In this work, we propose *diffusioN Optimized Sampling* (NOS), a guidance method for discrete diffusion models that follows gradients in the hidden states of the denoising network. NOS makes it possible to perform design directly in sequence space, circumventing significant limitations of structure-based methods, including scarce data and challenging inverse design. Moreover, we use NOS to generalize LaMBO, a Bayesian optimization procedure for sequence design that facilitates multiple objectives and edit-based constraints. The resulting method, *LaMBO-2*, enables discrete diffusions and stronger performance with limited edits through a novel application of saliency maps. We apply LaMBO-2 to a real-world protein design task, optimizing antibodies for higher expression yield and binding affinity to several therapeutic targets under locality and developability constraints, attaining a 99% expression rate and 40% binding rate in exploratory *in vitro* experiments.

## 1 Introduction

Optimizing protein sequences for improved function has the potential for widespread impact [63]. Among many potential applications in engineering and medicine, engineered antibodies can be used to create cancer therapeutics that are much less harmful to the patient than radiotherapy or chemotherapy. Because the space of possible proteins is vast and discrete, and experimental validation is slow and expensive, all practical methods for protein design must restrict themselves to a small enriched library of candidates to find a viable option in as few measurements as possible [44]. In practice these enriched libraries are usually obtained through massive high-throughput *in vitro* screening [67], or in the case of antibodies by injecting a live animal with the target antigen and sequencing the animal's immune response [52]. Generative protein models offer the tantalizing prospect of enriched libraries produced nearly instantly at a fraction of the cost. Success in real-world applications, however, has proven elusive, in part because naive generative models produce outputs that are similar to their training data and are therefore unlikely to improve target qualities [53].

There are many approaches to guided generation of proteins, but one broad and important distinction is between methods that search in sequence space and those that search in structure space. A basic tenet of molecular biology is "sequence determines structure, structure determines function" [9]. Hence when optimizing a protein for a desired function, it may seem more direct to design the protein in structure space, where gradient-based sampling methods can be used in tandem with carefully

---

*Equal contribution. [1]New York University, {[2]Prescient Design, [3]Antibody Engineering} Genentech.

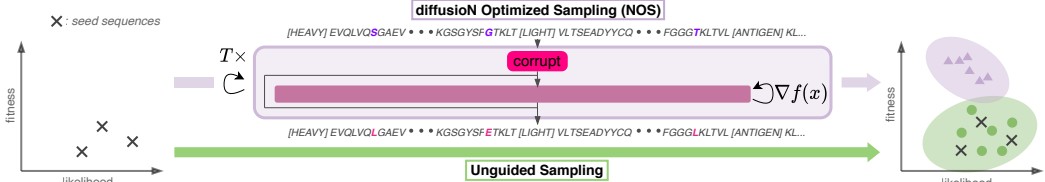

Figure 1: We propose diffusioN Optimized Sampling (**NOS**), a method for gradient-guided sampling from discrete diffusion models. NOS uses $T$ sampling steps of denoising diffusion, where each step consists of applying a corruption, gradient steps to optimize a value function, $f$, and sampling of the next discrete sequence, or corresponding latent state. NOS generates samples that optimize an arbitrary objective while maintaining high likelihood with respect to a reference distribution of sequences. We combine NOS with LaMBO, a strong Bayesian optimization method for sequence design [73], to make LaMBO-2, our improved method for protein design.

engineered potentials [1, 45, 77]. One of the drawbacks of this approach is the optimized structure must still be converted back to an amino acid sequence in order to be synthesized, a task known as "inverse-folding" [19]. There is no guarantee that the optimized structure can be realized by an actual sequence, and the inverse-folding procedure may not find the sequence if it exists. Structural models are also computationally intensive and limited by the scarcity of high-quality structural data. Searching directly in sequence space eliminates the need to recover sequence from structure. Protein sequence models are also comparatively fast, especially during inference, and can leverage sequence datasets that are often several orders of magnitude larger than their structural equivalents.

Although sequence models are arguably the most practical foundation for protein design, they have historically suffered from the challenges of optimizing discrete sequences, where gradient-based sampling is not directly applicable. As a result, many sequence search methods resort to gradient-free sampling methods like Metropolis-Hastings MCMC [78, 37], which are flexible but computationally expensive, eroding a key advantage over structure search. Several methods have been proposed that maintain gradient-based search by performing guidance in a continuous latent space, with a learned decoder to sample discrete sequences [33, 32]. Notably, Stanton et al. [73] proposed LaMBO (**La**tent **M**ulti-Objective **B**ayesian **O**ptimization), an optimization method that uses masked language model (MLM) style denoising guided with Bayesian acquisition values to address the online, multi-objective nature of real-world protein design. While LaMBO can quickly sample sequences with improved acquisition value, it has two key limitations. First, LaMBO is built on top of MLMs which, while strong representation learners are not strong generative models. In particular, MLMs lag behind other methods in producing high likelihood samples or infills. Second, despite being designed to improve known sequences instead of designing them completely from scratch, LaMBO and related methods have no principled framework for simultaneously enforcing an edit budget and choosing optimal edit locations based on that budget.

To address the first issue we propose **NOS** (diffusioN Optimized Sampling), a new method for controllable categorical diffusion (Figure 1). Diffusion models capture complex distributional dependencies by making iterative denoising steps, but there is relatively little previous work on how to control these processes. NOS generates sequences with both high likelihood and desirable qualities by taking many alternating steps between corruption, guidance, and denoising in the continuous latent space of the model. Our *in silico* validation shows that NOS outperforms many state-of-the-art structure and sequence-based baselines on both unguided and guided infilling tasks.[2] To address the second problem (choosing optimal edit locations) we propose using embedding-gradient feature attributions (i.e. saliency maps) to determine which positions on the sequence are most important to edit to improve the guidance objective value. We combine NOS with saliency-selected edits to create LaMBO-2, a more powerful variant of the original LaMBO algorithm. Exploratory *in vitro* experimental validation of our designs provides evidence that LaMBO-2 can be used to create enriched antibody libraries without the aid of additional *in vitro* high-throughput screening.

---

[2]https://github.com/ngruver/NOS

## 2 Related Work

**Discrete diffusions** Austin et al. [4] and Hoogeboom et al. [40] constructed diffusion models for discrete categorical data using a categorical noise process. Recently, categorical diffusion has shown promise as a competitor to autoregressive models in text generation for machine translation and summarization. The approaches can be roughly grouped into methods that apply categorical noise distributions directly to sequences (CMLM [31], SUNDAE [65]), and those that apply Gaussian corruptions to token-vector embeddings (SED [74], CDCD [21]). In this work we show that NOS can guide both types of categorical diffusions. Within the space of protein design, our method is also closely related to joint diffusion models over both sequence and structure [2, 51], which also circumvent inverse folding. Because these models still rely on structure information at training time, they can be limited by data availability in the same manner as pure structure models.

**Discrete generative guidance** Gradient guidance typically augments sampling from a generative model with gradient steps to increase the occurrence of a desired attribute [54]. Gradient guidance is natural within the framework of continuous diffusion models [20], and Li et al. [47] use this connection to perform gradient-guided sampling from a diffusion language model. To obtain a continuous space, they perform Gaussian diffusion [39] on word embeddings, decoding out to tokens using a linear head. The original method required many careful engineering interventions, e.g. clamping latent representations to the nearest word embedding, that have been improved by recent methods, such as CDCD [21], but gradient guidance has not been discussed for these more recent formulations.

To achieve a similar form of gradient guidance without carefully engineering a latent space, Dathathri et al. [17] and Yang and Klein [83] propose gradient-guided autoregressive models by using the decoder's activations as a gradient-friendly latent space. These methods alternate between sampling from logits and ascending the likelihood of a separately trained classifier model. Surprisingly, despite work on gradient guidance for continuous noise diffusions and autoregressive language models, there has been little work on gradient guidance for general categorical diffusions that predict denoised categorical distributions (e.g. CMLM, SUNDAE, CDCD), which is a topic we explore in detail. One closely related method proposed in the context of generative models of small molecules is DiGress [79], which performs gradient guidance on one-hot token embeddings to bias the categorical sampling distribution of a denoising model. In our setting we show that categorical and Gaussian discrete diffusions guided with NOS outperform PPLM and DiGress (Subsec. 5.2).

**Genetic algorithms** Evolutionary algorithms are a popular solution for black-box optimization in discrete spaces [3, 22]. These methods are often evaluated on their ability to optimize *in silico* proxy estimates of actual *in vitro* fitness, e.g. deep learning models trained on experimental datasets. In Subsec. 5.3 we baseline NOS against two genetic optimizers from protein design literature, AdaLead [70] and Proximal Exploration (PEX) [61]. We show these baselines rapidly degrade sequence likelihood as the proxy fitness is improved, limiting the effective number of edits that can be made before checking the actual fitness of the samples, ultimately limiting their sample efficiency and rate of convergence to optimal solutions. By contrast, NOS consistently improves proxy fitness while maintaining sequence likelihood.

## 3 Background

We pose protein design as the problem of finding sequences, $w \in \mathcal{A}^L$ with alphabet $\mathcal{A}$ and fixed length $L$,[3] which maximize a single objective $f(w)$ (e.g., binding affinity) or multiple objectives $f_1(w), \ldots, f_k(w)$ (e.g., expression yield, binding affinity, and aggregation tendency). Designs can be generated from random noise (*ab initio* design) or by making a fixed number of edits $B \in \{1, \ldots, L-1\}$ to a seed sequence $s \in \mathcal{A}^L$. A protein is only useful if it can be synthesized (i.e. expressed), and the objective value of non-expressing proteins is undefined since their properties cannot be measured. Therefore we must introduce the constraint $w \in \mathcal{E} \subset \mathcal{A}^L$, where $\mathcal{E}$ is the set of all expressible proteins. Since naturally occurring sequences must express in order to be observed, $p(w)$, the likelihood of a protein with respect to an empirical distribution of natural protein sequences, is often taken as a proxy for the tendency of a protein to express. In protein design, these proxies are

---

[3]Length change is enabled by the use of protein sequence alignments, which introduce a gap token "-".

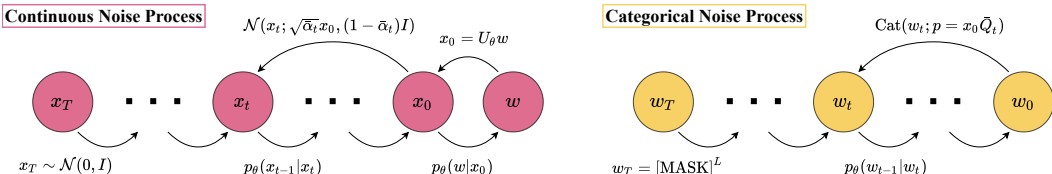

Figure 2: Two approaches to diffusion generative modeling for categorical variables. (**Left**) Categorical data is embedded into continuous variables with an accompanying continuous noise process. (**Right**) Categorical noise is applied directly to sequences, and corrupted sequences are denoised using standard language modeling methods.

typically called metrics of *naturalness*. Since we are looking for sequences that by definition have not yet been identified in nature, naturalness and our other objectives are often in tension.

We can trade off naturalness and objective value by drawing samples from the unnormalized density

$$\tilde{p}(w) = \exp(-\tilde{E}(w))/Z, \quad \tilde{E}(w) = E(w) - v(w), , \tag{1}$$

where $E(w) = -\log p(w)$ is a scalar energy function, and the value function $v : \mathcal{A}^L \to \mathbb{R}$ expresses the utility of a sequence with respect to our objectives. When designing proteins from primary sequence, sampling efficiently from the resulting energy function can be challenging. Simple approaches, such as the MCMC sampler used by Verkuil et al. [78] can require hundreds of thousands of steps to converge (Appendix C.2). Guided diffusion models are an appealing alternative because they construct a fixed-length Markov chain that quickly generates low-energy samples.

**Diffusion models** Denoising diffusion models construct samples by reversing a diffusion process that maps clean data points, $x_0$, to samples from a prior $\pi(x)$ (Figure 2). The forward process $(x_0 \to x_T)$ is composed of conditional distributions $p(x_t|x_{t-1})$ (i.e., the noise process) that admit closed forms for the conditional distributions $p(x_t|x_0)$ and $p(x_{t-1}|x_t, x_0)$ (e.g., independent Gaussian corruption). The reverse process $(x_T \to x_0)$ converts samples from the prior into samples from the learned data distribution $p_\theta(x_0)$ by repeatedly predicting the denoised variable $\hat{x}_0$ from noisy values $x_t$ and using the conditional distribution $p(x_{t-1}|x_t, \hat{x}_0)$ to derive a transition distribution, $p_\theta(x_{t-1}|x_t)$. The specific choice of noise process has been shown to significantly affect the likelihood and quality of image samples [71]. For categorical data there are two common approaches to constructing a diffusion generative model, depending on the nature of the noise process. We include brief descriptions below and a more detailed account in Appendix A.

**Continuous noise** To learn a distribution $p(w)$, one strategy is to first embed $w$ to a continuous variable $x_0$ with embedding matrix $U_\theta$ and apply Gaussian noise [21]. The prior is taken to be $\pi(x) = \mathcal{N}(0, I)$ while the forward process is $p(x_t|x_0) = \mathcal{N}(x_t; \sqrt{\bar{\alpha}_t}x_0, (1 - \bar{\alpha}_t)I)$ for $\bar{\alpha}_t \in [0, 1]$. The values of $\bar{\alpha}_t$ are determined by a user-specified corruption schedule. For the reverse process, we learn a function, $p_\theta(\hat{w}|x_t, t)$, to predict the sequence from noised points $x_t$ by minimizing the following loss with respect to $\theta$:

$$L(\theta) = \mathbb{E}_{w_0,t} \left[ -\log p_\theta(w_0|x_t) \right], \ x_t \sim p(x_t|x_0 = U_\theta w_0).$$

Using $p_\theta(\hat{w}|x_t, t)$ we can construct a distribution for the reverse process

$$p_\theta(x_{t-1}|x_t) = \sum_{\hat{w}} p\left(x_{t-1}|x_t, \hat{x}_0 = U_\theta \hat{w}\right) p(\hat{w}|x_t, t), \tag{2}$$

where $p(x_{t-1}|x_t, x_0)$ is also a Gaussian distribution. At inference time, we can use the learned reverse process to convert samples from $\pi(x)$ into samples from the learned distribution $p_\theta(x_0)$ by repeatedly sampling $p_\theta(x_{t-1}|x_t)$, followed by sampling $w \sim p_\theta(\hat{w}|x_0, 0)$.

**Categorical noise** Alternatively, Austin et al. [4] proposed a forward process which operates directly on $w$, by introducing an absorbing state for each token $w^{(i)} = [\text{MASK}]$. The forward process $(w_0 \to w_T)$ is defined by a discrete transition matrix, describing the probability of mutating a token into a [MASK], and the corresponding prior is simply a point mass on the sequence of all [MASK] tokens. To learn the parameters of the denoiser $p_\theta(\hat{w}_0|w_t, t)$ we maximize the likelihood of the denoising process on ground truth sequences

$$L(\theta) = \mathbb{E}_{w_0,t} \left[ -\log p_\theta(w_0|w_t) \right], \ w_t \sim p(w_t|w_0)$$

Then, as above, we can use the denoiser to construct the reverse process

$$p_\theta(w_{t-1}|w_t) = \sum_{\hat{w}_0} p(w_{t-1}|w_t, \hat{w}_0) p_\theta(\hat{w}_0|w_t, t) \tag{3}$$

where $p(w_{t-1}|w_t, w_0)$ is also a categorical distribution derived using Bayes' rule. To sample, the transition distribution is applied for $t = [T, ..., 0]$.

## 4 Methods

Now we present practical methods for efficiently sampling from $\tilde{p}(w) \propto p(w) \exp(v(w))$ (Eq. 1) by modifying the learned transition distribution with a learned value function $v_\theta(w)$. We then show how this sampling method can be used to perform protein design through guided infilling in sequence space. As before, we provide the most salient information below and the full details in Appendix B.

### 4.1 NOS: diffusioN Optimized Sampling

We introduce a general form of gradient guidance (NOS) for discrete diffusions with categorical denoising models, i.e. diffusion models that predict logits over the ground truth tokens (e.g. [21, 4]). The key challenge in applying gradient guidance to categorical data is simply the lack of a continuous representation. Fortunately, in any denoising network, e.g. $p_\theta(\hat{w}|x_t, t)$, the discrete sequence $w_t$ has many corresponding continuous representations in the form of hidden states of the model $h_t = g_d(w_t)$ for $d \in \{0, \ldots, D\}$, where $D$ is the depth of the encoder network and $g_0(w_t) = U_\theta w_t$. Notably, for the Gaussian diffusion models in Sec. 3, we can equivalently have $x_t = g_0(w_t)$, as corruption and sampling are performed on the learned token embeddings. In the case of the categorical noise diffusion $p_\theta(\hat{w}_0|w_t) = p_\theta(\hat{w}_0|h_t)$, and thus for the purpose of guidance, we can consider a general $p_\theta(\hat{w}|h_t)$ for both forms of corruption.

To sample from $\tilde{p}_\theta(w_t) \propto p_\theta(w_t) \exp(v_\theta(w_t))$, we construct a modified denoising model,

$$\tilde{p}_\theta(\hat{w}|h_t) \propto p_\theta(\hat{w}|h_t) \exp(v_\theta(h_t)).$$

This formulation requires that the denoising model and the value function share hidden states up to depth $d$, and that the value function also be trained on corrupted inputs $w_t$. In Appendix D.4 we propose a simple procedure for corrupted discriminative training inspired by label smoothing [76]. Using this modified denoising model we can construct modified transition distributions using Eq. 2 or Eq. 3. There is one key difference between these transition distributions: in the continuous case (Eq. 2), smooth steps are taken in the token embedding space, while in the discrete case (Eq. 3) the transition leads to large jumps from one token embedding to another. In either case, it is possible to sample a discrete sequence $w$ at any point in the chain using the logits of the denoiser $p_\theta(\hat{w}|h_t)$. When using Eq. 2 to derive a continuous transition distribution, we call the method **NOS-C**, and when using Eq. 3 for discrete transitions, we call the method **NOS-D**.

To sample from the modified transition distribution at each diffusion step, we use Langevin dynamics with temperature $\tau > 0$, with the update step,

$$h_t' \leftarrow h_t' - \eta \nabla_{h_t'}[\lambda \mathrm{KL}(p_\theta(\hat{w}|h_t')||p_\theta(\hat{w}|h_t)) - v_\theta(h_t')] + \sqrt{2\eta\tau}\varepsilon, \quad \varepsilon \sim \mathcal{N}(0, I), \tag{4}$$

where $\eta$ is the step size and $\lambda$ is the regularization strength, followed by sampling $p_\theta(w_{t-1}|h_t')$ or $p_\theta(h_{t-1}|h_t')$. While the gradient $\nabla_h v_\theta$ guides towards high values of the objective, the KL term ensures the resulting transition distribution still maximizes the likelihood of the original prediction.

NOS is related to the popular method plug-and-play language model (PPLM), which can be used for gradient-guidance of autoregressive language models [17]. PPLM guides sampling by taking gradient steps similar to Eq. 4 for each autoregressive decoding step (details in Appendix B). Unlike PPLM, NOS is a form of iterative refinement, meaning that tokens across the entire sequence can be modified at each optimization step. This distinction is particularly important for protein design, because function can be determined by complex interactions between distant parts of the sequence. As we see in Sec. 5, NOS leads to better trade-offs between likelihood and objective value.

### 4.2 LaMBO-2: function-guided protein design

Many unique challenges arise when applying guided diffusion to real-world protein design tasks. Our approach builds on the LaMBO-1 algorithm proposed by Stanton et al. [73], which explicitly

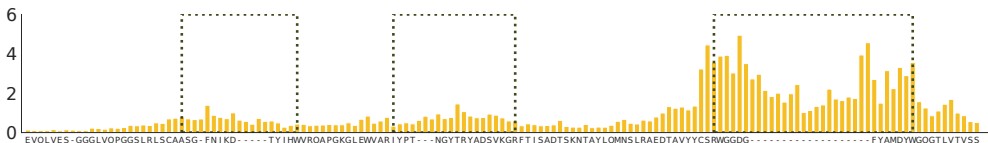

Figure 3: An example of a binding affinity saliency map produced by LaMBO-2 with NOS-D. For simplicity, only the variable heavy (VH) region of the hu4D5 antibody is shown. Positions corresponding to complementarity defining regions (CDRs) are enclosed in green boxes. Converting this saliency map to an edit position distribution will concentrate computational resources on editing CDRH3, which is often manually selected by experts. Some resources are also allocated to the framework and other CDRs since these positions may also affect binding.

accounts for the online, multi-objective nature of protein design by optimizing a multi-objective Bayesian acquisition function. LaMBO-2 replaces the guided MLM sampler with NOS, selects edit positions based on acquisition value saliency, and replaces the discriminative deep kernel Gaussian process (GP) with ensemble-based uncertainty quantification.

**Architecture and value function**  In order to apply the methods discussed in Subsec. 4.1 we require a generative diffusion model $p_\theta(w)$ and a discriminator $\hat{f}_\theta(w)$ which share hidden layers up to depth $d$. The discriminator is trained to predict the objective function $f$. Like LaMBO-1 our architecture consists of many task-specific feature extraction layers that share a bidirectional encoder. Bayesian optimization is an iterative cycle of inference and data acquisition. During the data acquisition phase of any iteration $i$ we need to find sequences with maximal *acquisition value* $v_i(w) = \mathbb{E}[u(w, f, \mathcal{D}_i)]$, where $\mathcal{D}_i$ is the data already available and the expectation is taken with respect to a posterior $p_\theta(f|\mathcal{D}_i)$ and $u$ is some utility function. For multi-objective tasks $u$ is the hypervolume improvement utility function [18], however we note that single-objective tasks are easily accommodated by a different choice of utility function [82]. To estimate the expectation we draw samples from $p_\theta(f|\mathcal{D})$ with an approach we call *partial deep ensembles*, where the discriminative layers of the model above the shared encoder are replicated $k$ times and randomly initialized [81]. We provide further details about partial deep ensembles and our learned discriminators in Appendix D.2 and D.3.

**Choosing edit positions**  When $B \ll L$ encoder-only architectures allow very precise control of edit positions since we will only change positions we corrupt. However, this feature introduces the need for some procedure to choose those positions, ideally where edits will most improve our objective value. We automatically select edit positions by computing the gradient of the value function with respect to $h_0$ to determine which positions affect the value estimate the most (see Figure 3 for an illustration). This method is related to the use of saliency maps to explain the decisions of classifiers [5, 69]. We use input saliency to induce a distribution over edit positions. Specifically, given an embedded sequence $h_0$ we define $s_i(h_0)$, the saliency with respect to $v$ of position $i \in \{1, \dots, L\}$ as

$$s_i(h_0) := \max\left\{ \left( \sum_{j=1}^{} \left| (\nabla_h v(h_0))_{ij} \right| \right)^{1/\tau}, \varepsilon \right\}, \quad \mathbb{P}[\text{edit } w_0^{(i)}] = \frac{s_i(h_0)}{\sum_j s_j(h_0)}, \tag{5}$$

where $\tau > 0$ is a temperature hyperparameter and $0 < \varepsilon \ll 1$. As $\tau \to +\infty$, $\mathbb{P}[\text{edit } w_0^{(i)}] = 1/L$ for all $i$. For each sequence we draw $B$ edit positions without replacement according to Eq. 5. We conserve parts of the input we cannot change (e.g. the antigen sequence) by setting the the saliency to 0 before computing the edit position distribution. Importantly, the diffusion sampling process can also preserve the original values of selected positions when appropriate. If we select a highly conserved position, then the predicted logits will be low entropy and the guidance will incur a large KL penalty for changes (Eq. 4).

# 5   Experiments

We evaluate our methods on three increasingly complex antibody design tasks. First we compare our trained diffusion models on unguided infilling tasks, showing that sequence diffusion methods

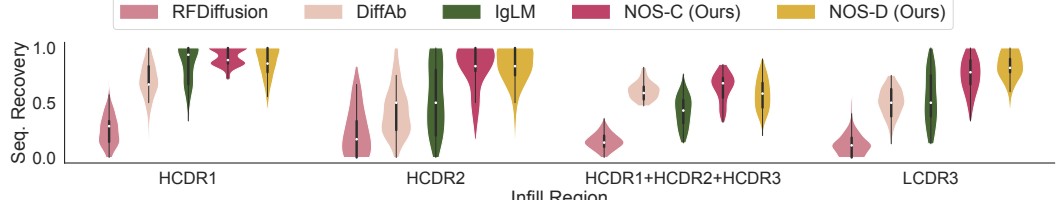

Figure 4: We infill antibody CDRs with discrete diffusion models (ours) and compare against structure-based diffusion models (DiffAb [51] and and RFDiffusion [80]) and an autoregressive antibody language model (IgLM [68]). We see diffusion on sequences alone–without structural priors–reliable leads to high sequence recovery. For structure based methods, we first fold seed sequences with IgFold [64] and then run joint sampling of sequence and structure for the CDR. We sample 10 infills for each of the 10 antibody seed sequences selected randomly from paired OAS [56].

consistently outperform structure-based methods when only predicted structures are available[4]. We then evaluate NOS by optimizing two objectives that can be simulated effectively *in silico*. Lastly, we evaluate LaMBO-2 on antibody lead optimization, with both *in silico* and *in vitro* experiments.

## 5.1 Unguided antibody CDR infilling

We focus on immunoglobulin G (IgG) format antibodies, which are comprised of a heavy (H) chain and a light (L) chain. Each chain has three complimentarity determining regions (CDRs), which tend to have strong effects on binding affinity to a target antigen but limited effects on other structural properties of the protein. Antibody design methods traditionally focus on proposing mutations to CDRs while leaving the rest of the protein fixed, which can be viewed as an infilling task. We select 10 seeds at random from paired OAS [56] and infill each CDR individually as well as in combination. To evaluate the performance of each model, we measure the sequence recovery rate, which is simply the accuracy of the infilling predictions given the ground truth sequence. As baselines, we include IgLM [68], a GPT2 language model trained on OAS, and two structure-based methods: DiffAb [51], a joint sequence-structure diffusion model trained on SAbDab, and RFDiffusion [80], a structural diffusion model trained on the PDB [10] that uses inverse folding to derive sequences. Although IgLM is trained with fill-in-the-middle augmentations [7], it does not natively support infilling multiple non-contiguous regions, and we do so by replacing regions that are not yet sampled with [UNK] tokens. For the structure-based methods, we provide starting structures generated with IgFold [64].

In Figure 4, we find that diffusion models often generate infills that are on-par or better than that those returned by IgLM by default, especially when multiple regions must be filled simultaneously. We also see that DiffAb, while being capable of sequence-structure co-design out of the box, often underperforms sequence-only diffusion, most likely because our sequence-based approaches have access to a larger training dataset, while paired datasets with sequences and structures are much more limited. Lastly RFDiffusion tends to generate relatively low likelihood CDR infills. The gap between DiffAb and RFDiffusion may be explained by the relative scarcity of antibody structures in the PDB compared to SAbDab, which has an antibody in every structure. The poor performance of structural methods on CDR infilling could also be a result of poor sequence recovery from structure during inverse folding, a problem that could be amplified for relatively unstructured loop regions like CDRs.

## 5.2 Optimizing antibodies for *in silico* objectives

To test guided sampling from our model, we run experiments on two simple single-objective tasks:

- The percentage of beta sheets, measured on primary sequence [15]
- The solvent accessible surface area (SASA) of the protein's predicted structure [64]

---

[4]In practical protein design campaigns it is infeasible to get ground truth structural measurements for proposed designs, and predicted structures are the only alternative available.

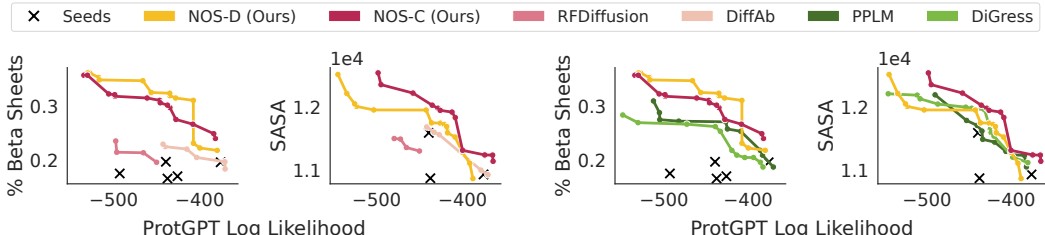

Figure 5: Comparing samples from NOS (ours) with alternative guided generation methods and structure-based models. NOS exhibits higher likelihood for similar or dramatically improved values of the objective. (**left**) Sequence diversification (resampling and selecting improved points) with DiffAb [51] or RFDiffusion [80]. DiffAb generates sequences and structures simultaneously, while sequences for RFDiffusion are obtained using ProteinMPNN [19]. Compared with NOS, these methods do not effectively optimize the objective and yield low-likelihood sequences. (**right**) Guided generation using PPLM [17], a guidance method for autoregressive language models (in this case IgLM [68]) and DiGress, a competing guidance method for discrete diffusion models [79]. NOS, PPLM, and DiGress are sampled for many settings of guidance strength (e.g. $\eta$ and $\lambda$ (Eq. 4)) to demonstrate the full range of trade-offs between the objective and likelihood. We provide details about hyperparameter settings in Appendix C.5 and additional density plots in Appendix C.6.

Since we want plausibly natural antibodies with high objective value we examine the Pareto front for samples optimized for each objective, with log-likelihood assigned by ProtGPT [29] (trained on Uniref50 [75]) plotted against the value of the objective. As an autoregressive guided baseline, we run PPLM, using IgLM as the base generative model (details in Appendix C.3). We use DiGress [79] as a guided diffusion baseline. DiGress uses gradients on one-hot representations to performing guidance in the embedding layer and is thus closely related to our approach. We discuss differences between the methods and the details of our DiGress experiments in Appendix C.5. For PPLM, DiGress, and NOS, we generate samples for many different setting of the control hyperparameters (Section 4.1), which yields samples across the spectrum from aggressively optimizing the objective to conservatively maintaining high likelihood. We also include DiffAb and RFDiffusion without guidance as baselines, as examples of popular "diversification" procedures, in which new samples are generated for later ranking and filtering. In Figure 5, we see that for both continuous and discrete corruptions NOS offers better trade-offs between optimizing the objective and maintaining high likelihood, while also generating high values of the objective at the extreme.

### 5.3 Antibody lead optimization: *in silico* evaluation

Having established the performance of NOS on simpler benchmarks, we now turn to real-world antibody design with LaMBO-2. From this point forward in all experiments we jointly condition on the heavy chain, light chain, and antigen sequence, and we jointly optimize the heavy and light chains only for improved expression yield and binding affinity to the antigen. As we discussed in Subsec. 4.2, one of the critical subproblems in Bayesian optimization is the identification of high value additions to the existing dataset. In this section we show that LaMBO-2 effectively applies NOS to this subproblem in the antibody design setting by finding high acquisition value sequences while preserving naturalness (which we quantify with the metric proposed by Shanehsazzadeh et al. [67]). We focus on optimizing hu4D5, a therapeutic antibody targeting the HER2 antigen[5]

**Comparison with genetic algorithms**  We first compare LaMBO-2 to two discrete optimization baselines, AdaLead and PEX. We generated 32 designs from the hu4D5 seed with each method, optimizing the same acquisition function derived from the same model. To ensure a fair comparison we limited all methods to a total of 512 model evaluations and a maximum of 2 edits per sampling iteration. We evaluated both the sample acquisition value and naturalness after each iteration. We identified an empirical naturalness threshold below which expression became unreliable and treated this threshold as a simple inequality constraint. Note that this experiment evaluates how each method balances the tradeoff between acquisition value and naturalness as sampling progresses, and does not involve evaluations of the actual black-box objective.

---

[5]HER2 is an important target for certain types of breast and stomach cancer [36].

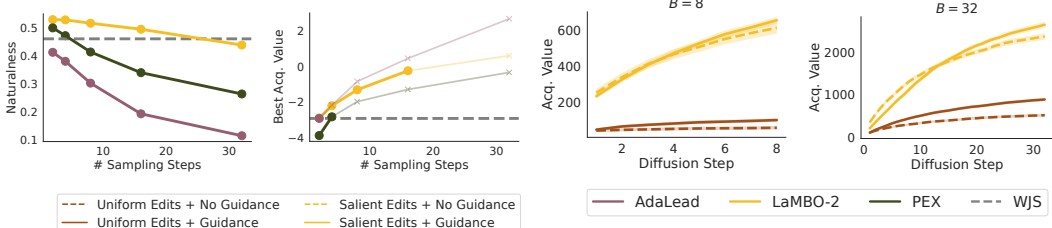

Figure 6: (**left**) Naturalness constraints present a challenge for genetic methods, which rapidly decline in naturalness even as their objective value improves. The grey dashed line is an empirical lower bound on naturalness above which *in vitro* expression is reliable. Although AdaLead and PEX both improve the acquisition value, they quickly leave well-supported areas of the search space (drop below the dashed line), shown by the faded section of each curve. By contrast, the naturalness of LaMBO-2 samples degrades much more slowly while consistently improving the acquisition value. (**right**) Ablating the effects of guidance and edit position selection. We start with the hu4D5 HER2 antibody and vary the edit budget $B \in \{8, 32\}$, optimizing for expression yield and binding affinity. For all choices of edit budget, we find that the effect size of edit position selection is much larger than that of guidance, making salient unguided edits a surprisingly strong baseline.

LaMBO-2 strictly dominates PEX in terms of naturalness and acquisition value at every sampling iteration, with PEX producing infeasible samples beyond 4 iterations. AdaLead improved sample value the most rapidly of all methods in this experiment, but also degrades naturalness the fastest, violating the constraint after only 2 sampling iterations. In contrast LaMBO-2 samples satisfy the naturalness constraint out to 16 sampling steps, producing the highest value feasible solutions. This result highlights the importance of accounting for distributional constraints when optimizing empirical proxies of fitness, since the quality of the proxy signal degrades rapidly outside the support of the training data. Genetic algorithms easily hack empirical models by leaving the support of natural sequences, where training data is necessarily absent, leading to poor quality solutions that nevertheless attain high acquisition value. In Appendix D.5 we show that both sequence and structure-based unguided infilling (i.e. random hit diversification) has the opposite behavior, producing samples with reasonable naturalness but low acquisition value.

**Effect of salient edits**    To separate and independently study the effects of guidance (NOS) and salient position selection, we present an ablation in Figure 6 for optimization with a relatively small edit budget $B$ ($B <= 10\%$ of mutable positions). To isolate the effects of salient edits we baseline against edit positions chosen uniformly at random, and to isolate the effects of guidance we set the step size $\eta$ (Eq. 4) to 0. Small edit distance constraints are common in antibody engineering because the goal is typically to increase binding affinity without altering the binding location on the antigen (i.e. the engineered antibody should bind to the same epitope) [43]. One heuristic way to constrain the design to the same epitope is to set $B \approx 8$, (about 2.7% of the antibody sequence length) [43], precisely the range we consider in Figure 6.

In the few edit regime we find that while both interventions improve sample objective value, selecting positions using saliency has a much larger effect than guidance. Although gradient guidance is a reliable and generally applicable tool for improved sampling, the scale of the edit position search dominates the scale of the search over token replacements that guidance affects. With a vocabulary of 21 tokens the number of possible token combinations ($21^8$) is dwarfed by the combinations of possible edit positions ($C_8^{300}$). Salient selection of edit positions is, therefore, key to any practical application of NOS in budget-constrained design. Interestingly, this facet of protein design differs significantly from guided sampling of images, where generation is typically limited to fixed locations [50, 14], not a fixed edit budget spread over any location that will optimize the objective. These additional degrees of freedom pose an extra challenge.

## 5.4  Antibody lead optimization: *in vitro* evaluation

As our final evaluation in Figure 7 we present results using LaMBO-2 (specifically with NOS-D) to optimize 20 seed antibodies distributed across 4 different therapeutic target antigens, including

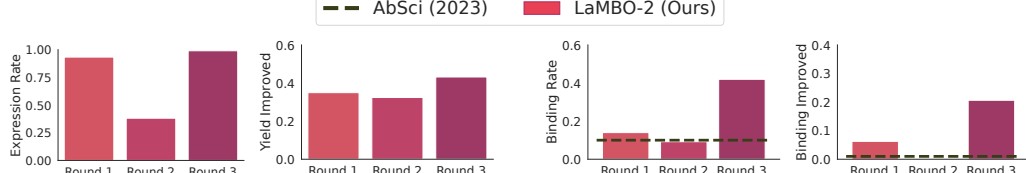

Figure 7: We use LaMBO-2 to optimize 20 seed antibodies for 4 different target antigens over three experimental rounds, retraining the model after each round. Some design choices and hyperparameters varied from round to round, with substantial impact on the results. In the last round we tested 56 antibodies and attained a 99% expression rate and 40% binding rate on average across targets. On average 43% of the expressing designs had higher yield and 21% of binding designs had higher binding affinity than the corresponding seed. These results are very encouraging when placed in context with a related experiment designing HER2 antibody libraries [67]. Our results provide evidence that enriched antibody libraries can be created *in silico* without the assistance of high-throughput *in vitro* screening.

hu4D5/HER2[6]. We tested 374 designs in total over three rounds, retraining the model after each round and varying a range of design choices and hyperparameters. While expression and binding performance varied from round to round across seeds and targets, by the final round we were able to generate multiple submicromolar binders for all 4 targets with a median of 5 edits to the seed. See Appendix D.7 for individual yield and affinity measurements and experimental details.

The improvements to yield and affinity over time can be attributed both to the data added to the training corpus and methodological insights gleaned after each round. For example, the sharp drop in expression in Round 2 can mainly be attributed to framework residue deletions that arose when $\lambda$ (the KL penalty coefficient) was set too small. In the following round we tried a range of larger $\lambda$ values and fixed the sequence lengths and expression immediately recovered.

Figure 7 also reports binding affinity results of a related experiment from Shanehsazzadeh et al. [67] for context, though we emphasize that there are substantial differences between our wetlab validation and that of Shanehsazzadeh et al. [67] which prevent a true apples-to-apples comparison. In the latter experiment 1M designs were generated for the HER2 target and screened with a high-throughput assay. After screening 421 designs were validated with a high-fidelity surface plasmon resonance (SPR) assay. In addition to wetlab screening, their experiment also restricted edits to specific antibody CDRs. We optimized antibodies for a range of targets including HER2 and relied exclusively on *in silico* screening before validating with SPR, while placing no explicit restrictions on the edit locations. Despite these differences, our results provide initial evidence that it is possible to generate enriched libraries of antibody designs exclusively with *in silico* methods operating only on primary sequence. While the experimental validation provided is preliminary, we are actively pursuing more rigorous experimental testing in the form of up-scaled and repeated expression and binding experiments and specificity assessment.

## 6 Discussion

There are many exciting directions for future work. The original LaMBO algorithm was used to optimize small molecules in addition to proteins, and applying LaMBO-2 to small molecule design is a fruitful direction, as LaMBO-2's improvements are not protein-specific. While sequence alignments are a convenient solution to the length change problem in protein design, padding methods [47] or diffusion with a variable-length corruption process (e.g. [60]) will be needed for applications like small molecules which do not admit alignments. We are also eager to consider optimizing much longer sequences, such as gene perturbations [42], which can exceed 20K tokens in length and may necessitate the use of recent advancements such as implicit convolutions [35, 57, 59] or clever modifications of self-attention [16, 13, 48]. More general notions of guidance such as classifier-free guidance [38] for text or class-conditional generation [62, 12] are another intriguing direction, since some goals are difficult to express as black-box functions or constraints [49, 58].

---

[6]Due to the sensitive nature of the data, we do not disclose the other seeds or drug targets.

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

# Appendix

## Table of Contents

## A  Extended Background

In this section we provide full descriptions of the diffusion processes introduced in Sec. 3.

### A.1  Continuous noise diffusion

The forward process is defined by noise variances $\beta$. We use the cosine variance schedule from Nichol and Dhariwal [55]. For convenience we further define

$$\alpha_t = 1 - \beta_t, \ \bar{\alpha}_t = \prod_i^t \alpha_i$$

The forward process is defined by the conditional distributions

$$p(x_t|x_{t-1}) = \mathcal{N}(x_t; \sqrt{1 - \beta_t}x_{t-1}, \beta_t I)$$
$$p(x_t|x_0) = \mathcal{N}(x_t; \sqrt{\bar{\alpha}_t}x_0, (1 - \bar{\alpha}_t)I)$$
$$p(x_t|w) = \mathcal{N}(x_t; \sqrt{\bar{\alpha}_t}U_\theta w, (1 - \bar{\alpha}_t)I)$$

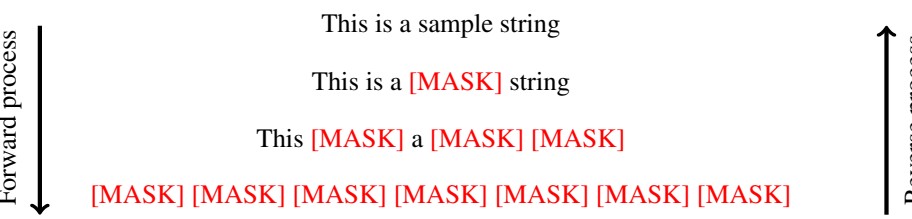

Figure 8: Illustration of a string gradually corrupted by [MASK] tokens.

where $U_\theta$ is an embedding matrix. The reverse process is defined by

$$\pi(x) = \mathcal{N}(0, I)$$

$$p(x_{t-1}|x_t, x_0) = \mathcal{N}\left(x_{t-1}; \mu_t, \sigma_t^2 I\right)$$

$$\mu_t = \frac{\sqrt{\bar{\alpha}_{t-1}}\beta_t}{1 - \bar{\alpha}_t}x_0 + \frac{\sqrt{\alpha_t}(1 - \bar{\alpha}_{t-1})}{1 - \bar{\alpha}_t}x_t$$

$$\sigma_t^2 = \frac{1 - \bar{\alpha}_{t-1}}{1 - \bar{\alpha}_t}\beta_t$$

$$p_\theta(w|x_t) = \text{Softmax}(\phi_\theta(x_0))$$

$$p_\theta(x_{t-1}|x_t) = \sum_{\hat{w}} p(x_{t-1}|x_t, x_0 = U_\theta\hat{w})p_\theta(\hat{w}|x_t)$$

### A.2 Categorical noise diffusion

Following Austin et al. [4] we define the MLM style categorical diffusion using transition matrices

$$[Q_t]_{ij} = \begin{cases} 1 & \text{if } i = j = m \\ \alpha_t & \text{if } j = m, i \neq m \\ 1 - \alpha_t & \text{if } i = j \neq m \end{cases}$$

and $\bar{Q}_t = Q_1 Q_2 ... Q_t$ for noise schedule $\overline{\alpha}_t \in [0, 1]$ (see Figure 8 for an illustration). These transition matrices correspond to categorical conditional distributions

$$p(w_t|w_{t-1}) = \text{Cat}(w_t; p = w_{t-1}Q_t)$$

$$p(w_t|w_0) = \text{Cat}(w_t; p = w_0\bar{Q}_t)$$

The reverse process is defined by

$$\pi(w) = 1[w = [\text{MASK}]^L]$$

$$p(w_{t-1}|w_t, w_0) = \text{Cat}\left(w_{t-1}; p = \frac{w_t Q_t^\top \odot w_0 \bar{Q}_{t-1}^\top}{w_0 \bar{Q}_t w_t^\top}\right)$$

$$p_\theta(w_0|w_t) = \text{Softmax}(\phi_\theta(w_t))$$

$$p_\theta(w_{t-1}|w_t) = \sum_{\hat{w}_0} p(w_{t-1}|w_t, \hat{w}_0)p_\theta(\hat{w}_0|w_t)$$

## B Methodological Details

### B.1 Infilling algorithm

We sample infills using the procedure in Algorithm 1. The infill mask $P$ is constructed by setting the index of conserved residue equal to 1, in this case at every residue that is not included in set of CDR regions being infilled. We use the same algorithm to perform the guided infilling in Subsec. 5.2, where it is extended with a guidance Langevin sampling step.

**Algorithm 1** Infilling with categorical denoising diffusion model

---

**Inputs:** Denoiser $p_\theta(\hat{w}|x_t, t)$, corruption process $p(x_t|x_0)$, infilling mask $P$, and seed sequence $s$
**Returns:** Sample from $\tilde{p}(w) = p_\theta(w|P, s)\exp(f(w))$
$x_T \sim p(x_T)$
$s_T \sim p(s_T|s)$
$x_T \leftarrow (I - P^\top P)x_T + P^\top s_T$
**for** $t = T, \dots, 1$ **do**
$\quad \big| \quad p(x_{t-1}|x_t) \leftarrow \sum_{\hat{w}} p(x_{t-1}|x_t, \hat{w})p_\theta(\hat{w}|x_t, t)$
$\quad \big| \quad x_t \sim p(x_{t-1}|x_t)$
$\quad \big| \quad s_t \sim p(s_t|s)$
$\quad \big| \quad x_t \leftarrow (I - P^\top P)x_t + P^\top s_t$
**end**
$w \sim p_\theta(w|x_0)$
**return** $w$

---

## B.2 Hidden State Langevin Sampling

Design of molecules or images with generative models is often posed as the problem of sampling from a posterior distribution $p(x|a)$ given the unconditional distribution $p(x)$ and attribute model $p(a|x)$. Indeed, reinforcement learning, the design of good actions in an environment, can also be framed as posterior sampling where $p(a|x)$ is the probability that a given state or state-action pair is optimal [46]. Methods that employ posterior sampling of this form are often call "plug-and-play" because $p(a|x)$ and $p(x)$ need not share parameters and therefore users can mix and match different instantiations [54, 17, 34, 26]

The most common way to sample from the posterior $p(x|a) \propto p(a|x)p(x)$ is through Langevin sampling on the unnormalized joint density $\tilde{p}(a, x) = p(a|x)p(x)$, with sampling steps

$$x^{i+1} = x^i + \eta \nabla \log \tilde{p}(a, x) + \sqrt{2\eta}z^i, \quad z^i \sim \mathcal{N}(0, I)$$
$$= x^i + \eta \left( \nabla \log p(a|x) + \nabla \log p(x) \right) + \sqrt{2\eta}z^i, \quad z^i \sim \mathcal{N}(0, I)$$

When we work with generative models over continuous random variables that permit a likelihood (e.g. normalizing flows), score function (e.g. diffusions), or energy (e.g. EBMs) $\nabla \log p(x)$ has a natural interpretation and sampling can be performed with essentially vanilla Langevin sampling. In other cases where only a denoising function over continuous variables is available, authors have proposed approximate samplers using an approximation of the score function [54].

When we instead hope to sample from a posterior over discrete random variables constructing an analogy to the score function $\nabla \log p(x)$ is challenging, and prior work adopts a different approach of regularizing the conditional sampling distribution $p(w|a)$ with unconditional sampling $p(w)$ in order to maintain high likelihood [17]. In autoregressive models, $p(w)$ is broken down using the chain rule, $p(w_t|w_{<t})$ and thus the appropriate regularization is

$$\mathrm{KL}(p(w_t|w_{<t}) \,||\, p(w_t|w_{<t}, a)) \tag{6}$$

In our case, the distribution $p(w)$ is factorized by the transition distributions $p(w_t|w_{t-1})$ (or their continuous analogies in token embedding space), and we hope to sample from the perturbed transition

$$\tilde{p}(w_{t-1}|w_t) = p_\theta(w_{t-1}|w_t)\exp(v_\theta(w_t))$$

The correct regularization term in our case is thus

$$\mathrm{KL}(p(w_{t-1}|w_t) \,||\, p(w_{t-1}|w_t, a))$$

To put the pieces together, we first recognize that the denoising model $p_\theta(w_0|w_t)$ can be broken down into an language model head, $H_\theta$, and trunk, $T_\theta$, with

$$h_t = T_\theta(w_t)$$
$$p_\theta(w_0|w_t) = H_\theta(w_0|h_t)$$

We can then perform Langevin sampling on the hidden representations, initializing with $h_t$, as shown in Algorithm 2. In the experiments above we set $\lambda_3 = 0$, as we saw no noticable benefit from adding additional stochasticity. Importantly, sampling from $p(w_{t-1}|w_t)$ already introduces randomness into the reverse process.

**Algorithm 2** Guided diffusion sampler

---

**Inputs:** Denoiser $p_\theta(\hat{w}|x_t, t) = [T_\theta, H_\theta]$, value function $v_\theta$, and weights $\lambda_1, \lambda_2, \lambda_3$
**Returns:** Sample from $\tilde{p}(w) = p(w)\exp(f(w))$
$w_T = [\text{MASK}]^L$
**for** $t = T, \ldots, 1$ **do**
    $p(w_{t-1}|w_t) \leftarrow \sum_{\hat{w}} p(w_{t-1}|w_t, \hat{w})p_\theta(\hat{w}|w_t)$
    $h^0 \leftarrow T_\theta(w_t)$
    **for** $i = 0, \ldots, K-1$ **do**
        $z^i \sim \mathcal{N}(0, I)$
        $p_h \leftarrow \sum_{\hat{w}} p(w_{t-1}|w_t, \hat{w})H_\theta(\hat{w}|h^i)$
        $h^{i+1} \leftarrow h^i + \lambda_1 \nabla_h v_\theta(h^i) + \lambda_2 \nabla_h \text{KL}(p(w_{t-1}|w_t)||p_h) + \lambda_3 z^i$
    **end**
    $w_{t-1} \sim H_\theta(h^K)$
**end**
**return** $w_0$

---

## C   Infilling / NOS Guidance

All of our diffusion models are train on all paired heavy and light chain sequences from OAS [56] (pOAS) combined with all sequences from SAbDab [25], aligned with ANARCI [24].

### C.1   Infilling experiment

For our trained diffusion models, we use Algorithm 1 without guidance, generating $P$ based on the indicated CDRs, using chothia numbering for consistency with DiffAb. For the baselines, we constructed wrapper scripts to convert the chosen CDR ids into each method's native format.

### C.2   MCMC comparison

Following Verkuil et al. [78], we construct a Markov chain using uniform random mutations to map a sequence $w$ to a mutated sequence $w'$, using the following Metropolis-Hastings correction:

$$p(\text{accept } w'|w) = \min\left(1, \frac{\exp(-E(w')/T)}{\exp(-E(w)/T)}\right),$$

where $T > 0$ is a temperature hyperparameter. While this method has appealing theoretical properties, obtaining good samples from this Markov chain in practice requires hundreds of thousands of steps of burn-in.

In our experiment (Figure 9), we define the energy, $E$, by combining sequence level probabilities assigned by IgLM with a beta sheets objective function trained on IgLM's representations. We construct the energy as

$$E(w) = p_{\text{IGLM}}(w) + \lambda v_\theta(w),$$

We tune $\lambda$ to generate sequences with approximately 40% beta sheets. We also tune the NOS $\lambda$ parameter (Eq. 4) to produce approximately 40% beta sheets.

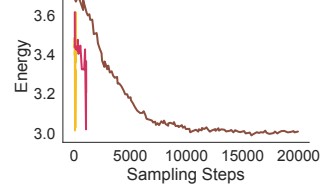

Figure 9: (**left**) Comparing convergence in sampling using a Metropolis Hastings-adjusted MCMC [78] against NOS models. Diffusion models (ours) accelerate sampling by two orders of magnitude while converging to similar energy values.

### C.3   PPLM details

In order to generate full (heavy and light chain) optimized antibodies with PPLM and IgLM, we train two separate value function models on IgLM's aggregated hidden representations, one for heavy chain sequences and one for light chain sequences. IgLM uses special tokens for both the chain identity and the species identity of each sequences, and we pass in appropriate corresponding tokens when calculating the hidden representations for each model. To determine the correct species token for each sequence, we use the predicted species returned by ANARCI [24]. Our value function is a

simple one-layer feed-forward neural network trained on top of the mean-aggregated representations for the corresponding chain identity.

To sample using PPLM, we overwrite the forward pass of the huggingface decoder used by IgLM to include a Langevin sampling step over the current hidden representations. We perform $K$ gradient steps to update the current hidden representation $\mathbf{h}'$ by descending on the objective

$$\lambda \text{KL}[p(\hat{w}|\mathbf{h}') \,||\, p(\hat{w}|\mathbf{h})] - v(\mathbf{h}')$$

where $h$ is the original hidden representation output by the model's encoder, and $\eta$ and $\lambda$ are the step size and regularization strength respectively. We ran optimization with both vanilla gradient descent and AdaGrad [23] and found AdaGrad to be more robust to poor specifications of the step size. For the results in Sec. 5, we draw samples and present results for all of the hyperparameter settings in Table 1

| $\lambda$ | 0, 0.001, 0.01, 0.1, 1.0 |
|---|---|
| $\eta$ | 0.5, 0.8, 1.1, 1.4, 1.7, 2. |
| $K$ | 5, 10 |
| optimizer | SGD, AdaGrad |

Table 1: Hyperparameter settings used for PPLM. $\lambda$ controls the strength of the regularization. Large values prevent sampling values that differ significantly from the unguided model. $\eta$ controls the size of steps taken in the latent space. Larger step sizes, when not too large, can increase the distance traveled in the latent space and the extent to which sampling can yield samples with high values of the objective.

One critical difference between controllable autoregressive models and controllable diffusions is the ability to resample previously sampled values. Procedures that allow for resampling are often called "iterative refinement" procedures because they can produce increasingly plausible generations by refining the model's previous output at each step in an iterative procedure. Because there are many potential differences between our NOS models and PPLM, including but not limited to the nature of iterative refinement, we performed an additional experiment to assess the impact of adapting a discrete diffusion to perform autoregressive sampling. Autoregressive models can themselves be thought of as diffusions with an idiosyncratic corruption process that masks out all tokens to the right of the last sampled token. As in our discrete corruption process, the prior is also a sequence of all mask tokens. Using this insight, we can run our trained discrete diffusions in autoregressive mode by contriving the sampling noise schedule to be autoregressive and recover an approximation of the timestep post-hoc from the percentage of masks at each step in autoregressive sampling.

Figure 10 shows the difference in objective values and likelihood for samples obtained by running the model in typical diffusion mode (iterative refinement) or in contrived autoregressive mode. We can see that on the beta sheets objective, iterative refinement has a noticeable positive impact on the objective values of the sample. This effect is also present in the SASA objective, but to a much more limited extent. We speculate that the iterative refinement facet of NOS is helpful for outperforming other methods but not completely sufficient.

### C.4   Model Architecture and Training

The gaussian and categorical diffusions are trained with the bert-small transformer backbone introduced by Bhargava et al. [8]. We use a cosine noise schedule for both diffusions and train for 100 epochs with a batch size of 64, optimizing with AdamW using an initial learning rate of 5e-3 with a linear warmup. The value function is a feed-forward neural network with one hidden layer. The value function is trained jointly with the denoiser by alternating optimization steps, with 5 steps on the generative objective for each step on the discriminative objective. We train the models for 100 epochs in total.

### C.5   Hyperparameter settings

For each guided sampling experiment with NOS, we sample using many different hyperparameter combinations in order to generate both conservative and aggressive optimization of the value function.

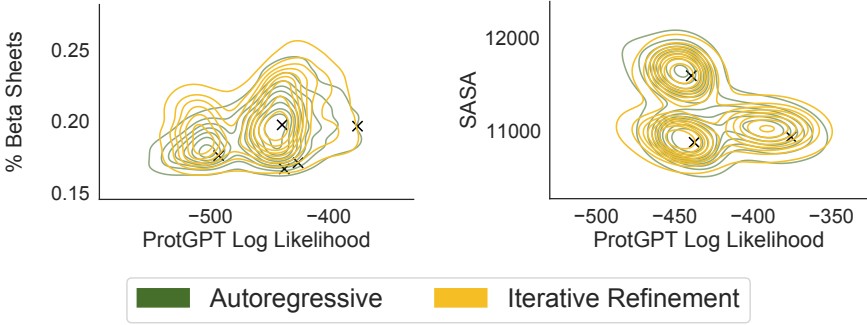

Figure 10: We compare samples from running our guided discrete diffusion (NOS-D) with diffusion style sampling versus autoregressive style sampling. We find that using an iterative refinement procedure does lead to consistent improvements in the objective value, though not to an extent that would suggest iterative refinement is sufficient for strong sampling performance.

The full hyperparameter settings for both objectives (beta sheets and SASA) and both corruption types (NOS-D and NOS-C) are shown in Table 2. In Table 2, there is an additional hyperparameter, "guidance layer", which we did not discuss at length in the main text of the paper. This parameter dictates whether we perform guidance in the first layer of the neural network (the token embeddings), as is standard in continuous diffusion models for discrete sequences, or the final layer of the neural network (the layer before the final linear head). In either case, we can use the same gradient descent objective and corruption process in each case and need only change the variable we propagate gradient updates to. Table 2 shows the hyperparameters used in the just Figure 5.

To aid intuition for the effects of each hyperparameter, we show the sample densities that result from each combination of $\lambda$ and $\eta$ in Table 2 when guiding in the first (Figure 11) and last (Figure 12) layer of the NOS-D and NOS-C models. We see that the most important parameter is $\lambda$, which controls how far samples tend to move from the seeds. We can also observe that guiding in the first hidden state tends to perform better when sampling with NOS-C, while guiding in the final hidden state tends to perform better with NOS-D.

**DiGress comparison** DiGress [79] is built on top of a model with one-hot encodings and discrete corruptions. The guided sampling procedure can be described as follows (using the notation from our submission): At each denoising step $t$, we use the one-hot encodings as a continuous variable and construct a perturbation distribution from a learned discriminative model $\hat{v} = v_\theta(w_t)$,

$$p_\theta(\hat{v}|w_{t-1}) \propto \exp(-\lambda\langle\nabla_{w_t} v_\theta(w_t), w_{t-1}\rangle)$$

We then sample the next value from the base diffusion transition $p_\theta(w_{t-1}|w_t)$ perturbed with $p_\theta(\hat{v}|w_{t-1})$,

$$w_{t-1} \sim p_\theta(w_{t-1}|w_t)p_\theta(\hat{v}|w_{t-1})$$

The key details for guided sampling can be found in the DiGress code repo, where we see that the guided distribution is the normalized product of the original denoising distribution and the softmax of the gradients scaled with $\lambda$.

On a theoretical level, this guidance has noticeably different properties from NOS. For large $\lambda$, the perturbation $p(w_t|\hat{v})$ collapses to a one-hot on the token index with the largest gradient value. For small values of $\lambda$, $p(w_t|\hat{v})$ becomes a uniform distribution. Therefore $\lambda$ interpolates $p(w_{t-1}|w_t)$ between the original unguided distribution and a one-hot in the max gradient direction. NOS also reduces to unguided infilling when $\lambda = 0$, but $\lambda > 0$ only modulates the direction of the gradient update. The distance between the guided and unguided distribution is controlled by the number of langevin steps and the step size hyperparameter $\eta$. Digress amounts to a single update step applied directly to the output token probabilities using a continuous relaxation of the one-hot encoded input, whereas NOS performs a sequence of local updates to hidden states that are actually continuous.

In our comparison, the embeddings and corruptions of each model are chosen to be:

1. NOS-C [Gaussian corruptions + learned embeddings]

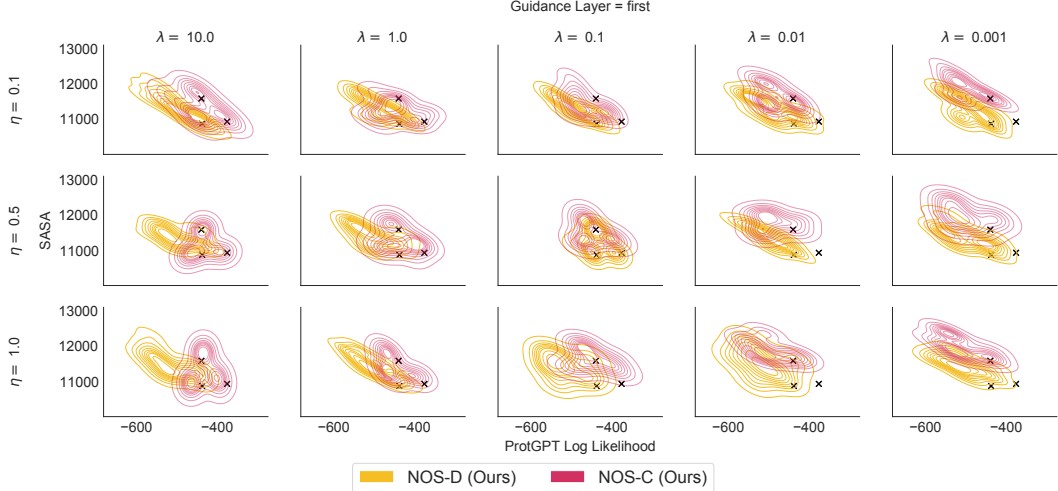

Figure 11: Density plots for every combination of the regularization ($\lambda$) and step-size ($\eta$) parameter, when performing guidance in the first layer (token embeddings) of the neural network denoiser. We observe that lambda has the strongest effect on trading off fitness under the objective with likelihood or closeness to the seed sequences.

2. NOS-D [Discrete (mask) corruptions + learned embeddings]

3. DiGress [Discrete (mask) corruptions + fixed one-hot encodings]

All models use the same backbone transformer and regression heads, facilitating an apples-to-apples comparison. For DiGress, we perform sampling for large range of scaling values $\lambda \in \{1e5, 3e4, 1e4, 3e3, 1e3, 3e2, 1e2, 3e1, 1e1, 1e0, 1e-1, 1e-2, 1e-3\}$. For each model, $\lambda$ modulates the degree to which the model prefers greedy sampling from the value function gradient.

| $\lambda$ | 0.001, 0.01, 0.1, 1.0, 10.0 |
|---|---|
| $\eta$ | 0.1, 0.5, 1.0 |
| $K$ | 5, 10 |
| guidance layer | first, last |
| optimizer | SGD, AdaGrad |

Table 2: NOS guided sampling hyperparameter settings. $\lambda$ controls the regularization strength, constraining the plausibility of samples, $\eta$, when chosen effectively, can effect the degree of optimization that takes place on the hidden states. The guidance layer is the layer in the neural network over which guidance is applied, the first being the token embeddings and the last being the final representations before the linear head. The same values are used for both NOS-D and NOS-C.

| $\lambda$ | 0, 0.001, 0.01, 0.1, 1.0, 10.0 |
|---|---|
| $\eta$ | 1.0 |
| $K$ | 10 |
| optimizer | AdaGrad |

Table 3: Hyperparameter settings used in Sec. 5. The guidance layer for NOS-D is final, and the guidance layer for NOS-C is last.

## C.6 Density plots

Because pareto fronts present only a partial view of sampling outcomes (focusing on the best case outcomes along each axis), we also include sample density plots to confirm that our methods consistently yield samples with better trade-off between likelihood and fitness. Figure 13 shows

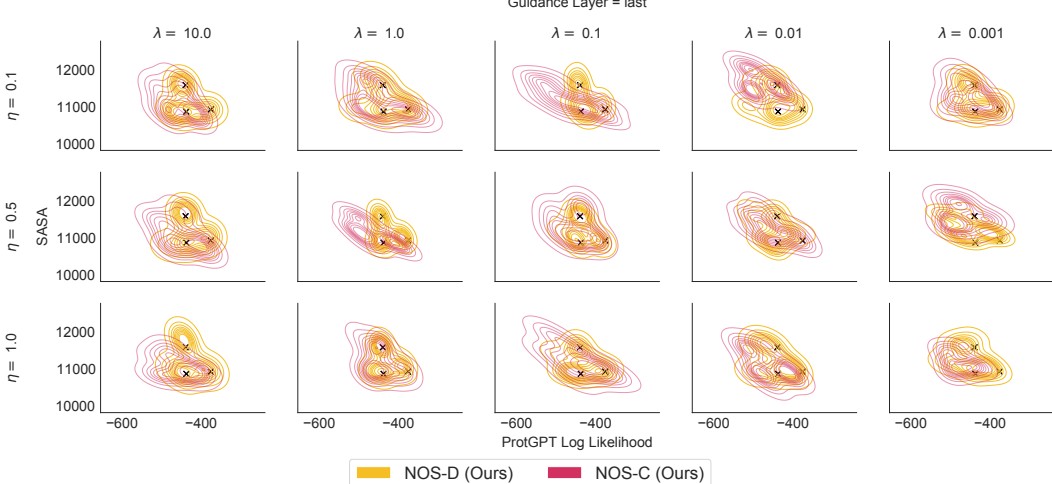

Figure 12: Density plots for every combination of the regularization ($\lambda$) and step-size ($\eta$) parameter, when performing guidance in the last layer (pre-logits layer) of the neural network denoiser. NOS-C and NOS-D exhibit quite different performance as a function of guiding the first or final hidden representation.

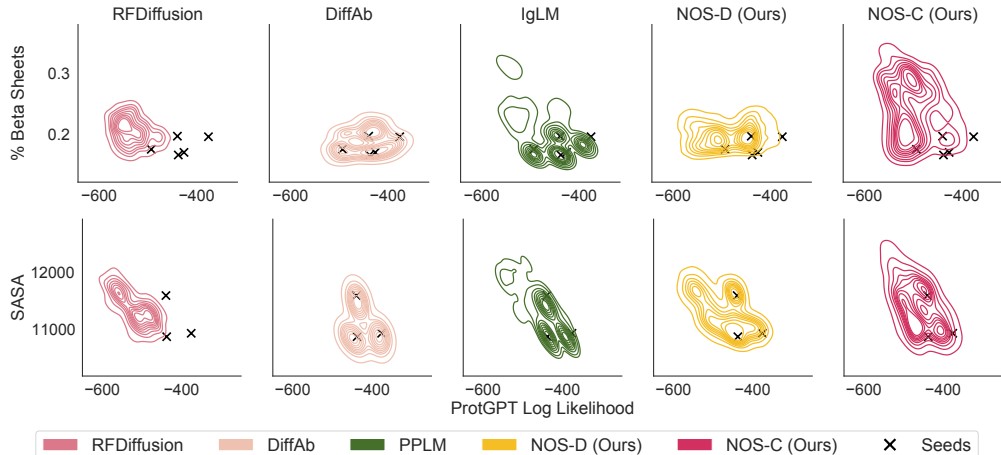

Figure 13: We compare sample densities for the methods presenting in Sec. 5, in order to augment the limitations of simply showing pareto fronts. We see that NOS-C and NOS-D can both consistently generate samples with favorable trade-offs while other methods tend to radically decrease likelihood with little benefit to the value function or be relatively limited to the neighborhood around the seed sequences.

density plots for NOS and baselines when optimizing each of the two objectives (percentage of beta sheets and SASA). We find that DiffAb and IgLM samples tend to cluster around the starting seeds, while RFDiffusion samples tend to generate more diverse samples under the objective, but often with much lower likelihood than the seed sequences. By contrast, both NOS methods consistently improve values of the objective without sacrificing likelihoods.

# D  LaMBO-2

## D.1  Intro to Multi-Objective Bayesian Optimization

When there are multiple objectives of interest, a single best (i.e. strictly dominant) sequence $\mathbf{x}^*$ may not exist. Suppose there are $k$ objectives, $f : \mathcal{X} \to \mathbb{R}^k$. The goal of multi-objective optimization

---

**Algorithm 3** LaMBO-2: one guided discrete diffusion step

---

**Inputs:** Seed sequence $w_0$, edit budget projection $P$, diffusion timestep $t$, corruption function $c(w, t)$, constraint function $u(w)$, encoder $g_\theta(w)$, value function $v_\theta(h)$, decoder $d_\theta(h)$, regularization strength $\lambda$, SGLD step-size $\eta$ and temperature $\tau$.

**Returns:** Best feasible sample from SGLD chain with distribution $p'(\mathbf{x}) \propto p(\mathbf{x}) \exp(f \circ g(\mathbf{x}))$

$w^*, v^* = w_0, v_\theta \circ g(w_0)$              (initialize optimal solution)

$w'_0 = c(w_0, t)$                             (apply diffusion noise)

$h'_0 = g_\theta(w'_0)$                           (initialize hidden state)

**for** $i = 1, \ldots, I$ **do**

    $\texttt{loss} = \lambda \mathrm{KL}[d_\theta(h'_{i-1}) || d_\theta(h'_0)] - (1 - \lambda) v_\theta(h'_{i-1})$

    $h'_i = h'_{i-1} - P(\eta \nabla_{h'} \texttt{loss} + \sqrt{2\eta\tau}\varepsilon), \quad \varepsilon \sim \mathcal{N}(\mathbf{0}, I)$      (projected SGLD step)

    $w_i \sim d_\theta(h'_i)$                          (decode hidden state)

    **if** $v^* < v_\theta \circ g_\theta(w_i)$ & $u(w_i)$ **then**

        $w^* \leftarrow w_i$

        $v^* \leftarrow v_\theta \circ g_\theta(w_i)$

    **end**

**end**

**return** $w^*, v^*$

---

(MOO) is to identify the set of *Pareto-optimal* (i.e. non-dominated) solutions such that improving one objective within the set leads to worsening another. We say that $\mathbf{x}$ dominates $\mathbf{x}'$, or $f(\mathbf{x}) > f(\mathbf{x}')$, if $f_j(\mathbf{x}) \geqslant f_j(\mathbf{x}')$ for all $j \in \{1, \ldots, m\}$ and $f_j(\mathbf{x}) > f_j(\mathbf{x}')$ for some $j$. The set of non-dominated solutions $\mathscr{X}^*$ is defined in terms of the Pareto frontier (PF) $\mathcal{P}^*$,

$$\mathscr{X}^\star = \{\mathbf{x} : f(\mathbf{x}) \in \mathcal{P}^\star\}, \quad \text{where } \mathcal{P}^\star = \{f(\mathbf{x}) : \mathbf{x} \in \mathcal{X}, \nexists \mathbf{x}' \in \mathcal{X} \text{ s.t. } f(\mathbf{x}') > f(\mathbf{x})\}. \quad (7)$$

MOO algorithms typically aim to identify a finite approximation to $\mathscr{X}^\star$ (which may be infinitely large), within a reasonable number of iterations. One way to measure the quality of an approximate PF $\mathcal{P}$ is to compute the hypervolume $\mathrm{HV}(\mathcal{P}|\mathbf{r}_{\mathrm{ref}})$ of the polytope bounded by $\mathcal{P} \cup \{\mathbf{r}_{\mathrm{ref}}\}$, where $\mathbf{r}_{\mathrm{ref}} \in \mathbb{R}^m$ is a user-specified *reference point*.

$$u_{\mathrm{EHVI}}(\mathbf{x}, f, \mathcal{D}) = \mathrm{HVI}(\mathcal{P}', \mathcal{P}|\mathbf{r}_{\mathrm{ref}}) = [\mathrm{HV}(\mathcal{P}'|\mathbf{r}_{\mathrm{ref}}) - \mathrm{HV}(\mathcal{P}|\mathbf{r}_{\mathrm{ref}})]_+, \quad (8)$$

where $\mathcal{P}' = \mathcal{P} \cup \{\hat{f}(\mathbf{x})\}$ [27, 28, 18]. To decide where to query $f$ next, we search for $\mathrm{argmax}_{\mathbf{x}} \mathbb{E}[u_{\mathrm{EHVI}}(\mathbf{x}, f, \mathcal{D})]$, where the expectation is w.r.t. $p(f|\mathcal{D})$.

### D.2 Discrete EHVI

Although expression yield and binding affinity are both continuous measurements, we chose to discretize them and model them as classification with a softmax likelihood (See Appendix D.4). As a result we needed an extension of EHVI for discrete outcomes. Informally, EHVI is simply computing the HVI for different realizations of $f$ and marginalizing $f$ using $p(f|\mathcal{D})$. Instead of taking $f$ to be the latent function of some regression $y = f(w) + \varepsilon$. $\varepsilon \sim \mathcal{N}(0, \sigma^2)$, we instead take $f$ to be the logits of a categorical distribution, $p(y = i|w, \mathcal{D}) = \int \texttt{softmax}_i(f(w))p(f|\mathcal{D})df$.

Let $\mathbf{y} = [y_1 \cdots y_k]^\top$. Given a set of baseline points $\mathcal{B} \subset \mathcal{A}^L$ we define $\mathcal{P}$ (Eq. 8) using the posterior mean $\hat{\mathbf{y}}(w) = \mathbb{E}[\mathbf{y}|w, \mathcal{D}]$, $w \in \mathcal{B}$. We model $y_1, \ldots, y_k$ as conditionally independent given some shared hidden state $h = g_d(w)$, so $p(\mathbf{y}|h, \mathcal{D})$ factorizes nicely. Finally we define $\mathcal{P}' = \mathcal{P} \cup \{\mathbf{y}\}$ and take the expectation of Eq. 8 w.r.t. $p(\mathbf{y}|h, \mathcal{D})$. Since $p(\mathbf{y}|h, \mathcal{D})$ is discrete and factorizes, we can marginalize in closed form when $K_1 \times \cdots \times K_k$ is not too large, where $K_i$ is the number of classes corresponding to the discretization of the original continuous $f_i$.

### D.3 Architecture and Hyperparameters

The inputs of the LaMBO-2 model for antibody design are the variable heavy (VH) and variable light (VL) regions of the antibody sequence as determined by Aho alignment with ANARCI, as well as the (unaligned) antigen sequence. Note that the concatenation of the antigen to the input makes the samples from the generative head conditional on the antigen as well as the unmasked portion of the antibody sequence. The LaMBO-2 model jointly predicts antigen-conditional categorical

token distributions for corrupted positions and discriminative distributions over protein properties. Discriminative predictions that should not depend on the antigen are made invariant through data augmentation with random antigen sequences. See Algorithm 3 for an overview of a single guided diffusion step with LaMBO-2.

**Model Architecture:** our architecture for this experiment is inspired by the one proposed by Stanton et al. [73]. In particular we jointly a train an encoder shared between a generative discrete diffusion head and discriminative heads which predict expression and affinity. Rather than use a deep kernel GP, we simply ensemble 10 heads for each discriminative task to obtain uncertainty estimates. Like Stanton et al. [73] for this experiment we use 1D CNN residual blocks (kernel width 9), with layer normalization and sinusoidal position embeddings. The shared encoder was comprised of 4 residual blocks, and each task head was comprised of 2 residual blocks followed by a linear layer, with the exception of the generative head which was just a linear layer on top of the shared embeddings. Note that in future work self-attention layers could be used instead of CNN layers, as was the case for the pOAS experiments in Sec. 5. We set the embedding dimension to 32, and the latent channel dimension to 256.

**Training Hyperparameters:** The LaMBO-2 model is both a jointly trained generative and discriminative model, as well as a true multi-task model, which is necessary since measurements for various protein properties are often missing from a substantial fraction of rows in real-world datasets. We trained for 500K gradient updates using the Adam optimizer with $\eta = $ 1e-3, $\beta_0 = 0.99, \beta_1 = 0.999$. At each gradient step we randomly sampled a task head and task minibatch (batch-size 121) and updated the corresponding weights (including shared weights). We used a linear learning rate warmup over 10K gradient updates, and decayed the learning rate to 1e-6 with a cosine schedule. We did not regularize with weight decay or dropout.

**Generation Hyperparameters:** to generate the designs in Figure 7, we sampled 1K designs from a pool of seed antibody sequences hand-selected by domain experts. For each seed we set the total edit budget shared between chains to $B = 16$. In this experiment each infilling method took 16 diffusion steps, using an inverse linear noise schedule $\overline{\alpha}_t = 1/(1 + t)$. Although the models were trained with a standard cosine noise schedule, we found the inverse linear schedule gave better results in terms of sample acquisition value at generation time. Within each diffusion step we took 64 Langevin steps, with noise scale $\tau = $ 1e-2. For guided infills with uniformly distributed edit positions we set $\tau = $ 1e6. For guided infills *with* saliency-informed edit position selection we set $\tau = 0.1$. We set $\lambda = 0.5$ to balance the tradeoff of sequence likelihood and value during guidance.

**Generation Constraints:** in addition to the edit budget locality constraint, our LaMBO-2 designs were also constrained to meet certain sequence liabilities constraints:

- **Canonical Cysteine Conservation:** there are specific conserved cysteine residues in antibody sequences which play a crucial role in the formation of disulfide bridges. Disulfide bridges are covalent bonds formed between two cysteine residues through oxidation of their sulfur atoms. These bridges contribute to the overall structural stability and integrity of antibodies.

- **No Unpaired Cysteines:** odd numbers of cysteines within individual chains (i.e. unpaired cysteines) are generally undesirable since they can lead to non-native disulfide bonds between different antibody molecules, which may disrupt assembly, folding, or function.

- **No Glycosylation Motifs:** A glycosylation motif is a specific amino acid sequence within a protein that serves as a recognition site for the attachment of sugar molecules. The presence of a glycosylation motif in a protein can affect its stability, solubility, activity, and function. The addition of sugar molecules can alter the protein's conformation, change its interactions with other proteins or molecules, and affect its trafficking and localization within the cell.

### D.4 Training Data, Class Imbalance, and Label Smoothing

**Training Data:** the expression task heads were trained on a dataset of 10K linear transfection expression measurements, which was subsequently augmented to 160K rows by pairing the same measurements with different random antigens to teach the model to ignore the antigen sequence when predicting expression. The binding task heads were trained on a dataset of 10K SPR affinity measurements for various antigens, which was then augmented to 12K rows by pairing binders with different random antigens and imputing a non-binding label. This augmentation is important for

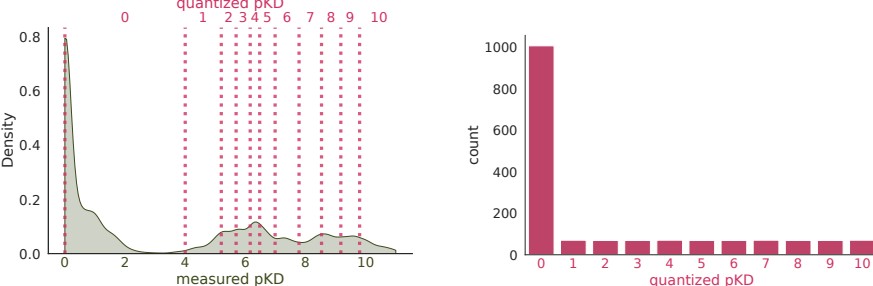

Figure 14: An illustration of using quantization to address heavily imbalanced data. On the right we show the original marginal label distribution in green, and the discretization boundaries as dotted lines. The boundaries are defined by a minimal level of affinity to be considered a binder ($pKD = 4$), and pKD deciles computed from the remaining measurements.

training a pan-target affinity model, since experimental measurements of affinity to off-target antigens are uncommon. Note that the expression and affinity data only partially overlapped, necessitating the multi-task architecture described in Appendix D.3. The generative diffusion head was trained only on binding antibody-antigen pairs in the SPR binding data.

We did not pretrain our LaMBO-2 models. It is likely that performance could be improved with the right pretraining corpus, however it is unclear if datasets like pOAS are particularly useful for pretraining antibody design models since most do not report antigen sequences and may not have the right level of variability. In any case, it is very encouraging to see positive real-world results before scaling in earnest.

**Label Discretization.** As noted above, biological data tends to be very imbalanced, and historical experimental data even more so since there are strong selection effects imposed by the scientists collecting the data. We chose to discretize continuous properties like expression yield and binding affinity, making it easier to correct for class imbalance by upsampling minority classes. In Figure 14 we illustrate our discretization scheme. Any antibody-antigen pair with $-\log(KD)$ (pKD) less than 4 was assigned to the non-binding class 0. Then binders were assigned to classes 1 - 10 based on which pKD decile (computed from binders only) they resided in. One consequence of this scheme is increasing any objective value by one unit corresponds to moving up one decile in the empirical label distribution.

**Training Discriminators on Noisy Inputs:** the benefits of discretization are not limited to addressing class imbalance. Working with discretized labels also allowed a simple approach to training the discriminator on corrupted inputs inspired by label smoothing [76]. We train the discriminators with the same noise schedule as the diffusion model and the usual cross-entropy loss, using modified labels

$$\overline{\mathbf{y}}_t = \overline{\alpha}_t * \mathbf{y} + (1 - \overline{\alpha}_t)/K * \mathbf{1},$$

where $\mathbf{y}$ is the one-hot encoded label and $K$ is the number of classes. Informally, as $\overline{\alpha}_t \to 0$ the discriminator reverts to a uniform prior since the inputs are not distinguishable. Training on corrupted inputs avoids evaluating the value gradient on out-of-distribution inputs during generation, and causes the strength of the value gradient to grow as the diffusion progresses and the samples become more defined.

## D.5 Baselining LaMBO-2 Against Unguided Sequence and Structure-Based Diversification:

**Structure-Based Diversification** We have shown that we can effectively optimize antibodies for predicted yield and affinity, and our method performs well compared to unguided sequence-based infilling methods. We expand our evaluation for this task to include unguided infilling with DiffAb and RFDiffusion of CDRs H2 and H3 of hu4D5 (i.e. the seed), a publicly released therapeutic antibody that is ideally suited for structure-based methods since we have a ground truth crystal structure of hu4D5 docked with its target ERBB2. While it is not feasible to validate the resulting designs *in vitro* during the author response period, we can compare the AntiBERTy naturalness scores and the acquisition value (log expected hypervolume improvement or log-EHVI) of the designs relative to our guided infills (Fig. 15). To summarize, unguided structure-based infilling produces

high likelihood samples, but even when conditioned on the antigen the distribution shift toward better predicted function is very slight.

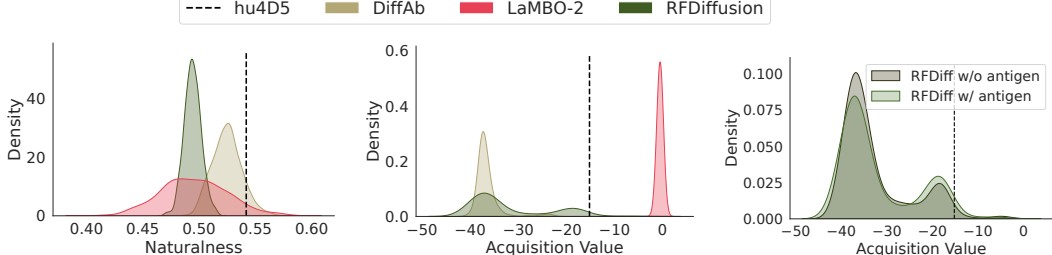

Figure 15: **(left)** we find that structure-based infills, particularly from DiffAb, tend to score consistently well on naturalness. Guided infilling produces a much wider range of scores, but the mode is very close to that of RFDiffusion. **(middle)** as assessed by the same model used to guide towards higher yield and binding affinity. The guided infills have very high acquisition value, since they were explicitly optimized for that outcome. Given 1024 samples each, DiffAb failed to produce any sequences of higher expected value than the seed, and RFDiffusion produced only 7 marginally improved designs. We also took the opportunity to assess the sensitivity of RFDiffusion to the antigen by comparing infills generated using the antibody stucture only **(right)**. While the effect is not large, antigen information does produce a small shift in the distribution of acquisition values to the right.

**Sequence Diversification** This *in silico* evaluation compares two variants of LaMBO-2 (one using NOS-C, the other NOS-D) against a competing method, walk-jump sampling (WJS), an unguided smoothed discrete sampling algorithm proposed by Frey et al. [30]. Each method generated 1K designs from the same set of seeds, and all methods were restricted to $B = 8$ edits. LaMBO-2 chose all edit positions automatically along the entire antibody sequence, whereas WJS was given manually selected edit positions restricted to CDRs only. In the left two panels of Figure 7 we compare the predicted expression yield, predicted binding affinity, and naturalness of the antibody designs, using the metric proposed by . Comparing the Pareto frontiers obtained from each set of designs, we see that while WJS excels at generating "natural" antibodies, it struggles to generate designs at the higher end of the objective range. Conversely LaMBO-2 designs (particularly those generated with NOS-C) have high predicted objective value but also lower naturalness scores. LaMBO-2 designs generated with NOS-D strike a balance between the two extremes.

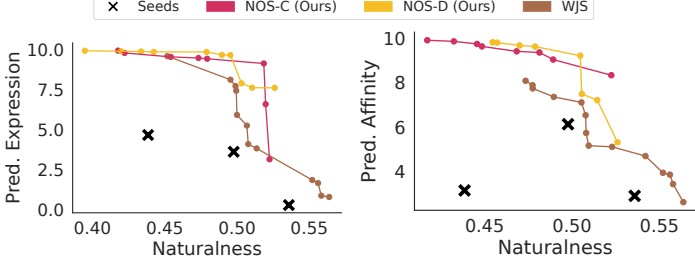

Figure 16: We evaluate LaMBO-2 in the context of real-world antibody lead optimization. LaMBO-2 can use either NOS-C or NOS-D to generate design libraries with higher predicted objective value than the unguided sampling baseline WJS [30], however intensive optimization comes at the cost of reduced naturalness (panels **left** and **center**).

### D.6 Are Saliency Maps Reliable?

There is substantial controversy regarding the reliability of input-gradient-based feature attribution methods, specifically related to their ability to consistently highlight ground truth task-discriminative features and ignore irrelevant features. For example, Hooker et al. [41] claim that random attribution is competitive with input-gradient methods, and Casper et al. [11] claim that gradient-free attribution outperforms input-gradient competitors. On the other hand, many papers claim that specific types

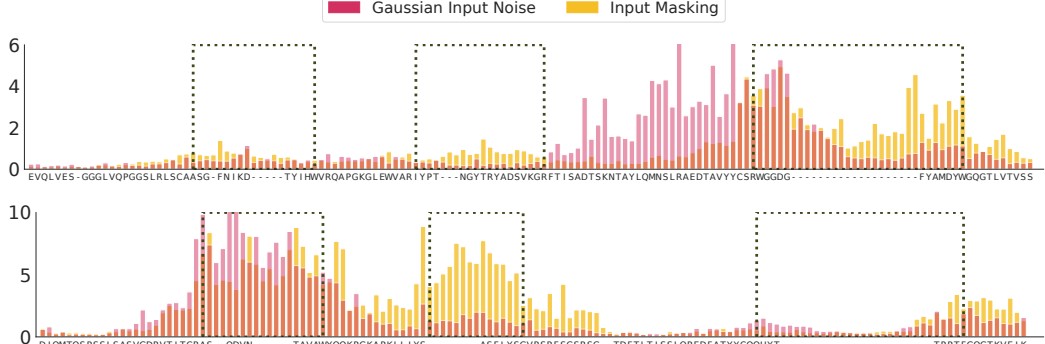

Figure 17: Binding affinity feature attributions for hu4D5 produced by independent models trained with different input corruptions. While the attributions do not match exactly, there is substantial agreement on the importance of CDRH3 (top panel) and CDRL1. Some importance is also assigned to various framework regions, which could be related to the fitness of different antibody germlines. We emphasize that these models were trained solely on aligned sequences, with no additional positional information.

of regularization can improve the performance of input-gradient attribution, including adversarial training [66], mask denoising [6], and model curvature penalties [72].

A thorough investigation of these claims is beyond the scope of this work, however we have found that saliency maps produced by independent models trained with different corruption processes seem to consistently highlight specific regions of the antibody sequence (Figure 17). It is also worth noting that most of the related literature evaluates feature attribution in the offline setting. In LaMBO-2 feature attributions are used online to intervene on the data collection process (specifically where to introduce changes in the antibody sequences). If LaMBO-2 changes a position that does not affect function it is reasonable to conjecture that input-gradient attributions would adjust accordingly after the model is retrained for the next round. Further investigation into feature attribution in decision-making contexts (as opposed to *post hoc* interpretability) is an exciting direction for future work.

## D.7 Wetlab Validation

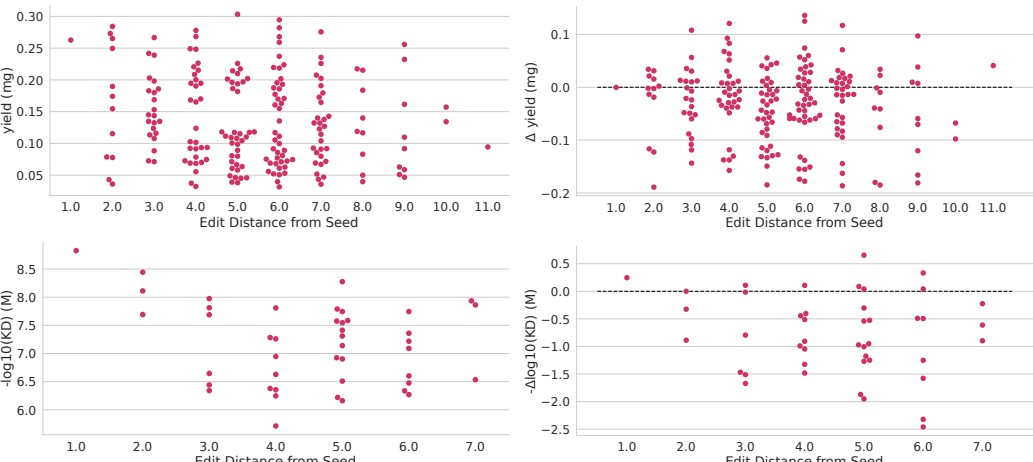

Figure 18: Here we show the experimentally validated yield **(top)** for all expressing designs and affinity **(bottom)** for all binding designs as a function of edit-distance from the original seed. In the right column we show the absolute measurement, and the left column shows the change relative to a seed measurement in the same batch.

In this section we briefly summarize the experimental procedures used to validate LaMBO-2 designs *in vitro*. Designed antibody sequences from LaMBO-2 were expressed and purified, and surface plasmon resonance (SPR) measurements were used to determine binding affinity. See Figure 18 for a plot of design binding affinity vs. edit distance from seed antibody.

**Plasmid Construction and Antibody Production:** synthesized DNA of antibody variable domains (Twist Biosciences) were cloned into mammalian expression vectors using Gibson assembly. The whole vector was amplified using PrimeStar Max polymerase (Takeda). PCR products were transfected transiently in 1mL Expi293 cell culture. Expression lasted 7 days before harvest. Antibodies were affinity purified over a MAb Select SuRe resin (Cytiva), and their concentration was measured by optical density at 280nM.

**Binding Affinity Measurements:** affinity of the antibodies towards their target antigen was measured by surface plasmon resonance (SPR) at 37 °C on a Biacore 8K instrument (Cytiva) in HBS-EP+ buffer (10 mM Hepes, pH 7.4, 150 mM NaCl, 0.3mM EDTA and 0.05% vol/vol Surfactant P20). Antibodies were captured on a Protein A chip and their target antigen were injected for 5 minutes and allowed to dissociate for 10 minutes at 30ul/min. The surface was regenerated between cycles with 10 mM glycine pH 1.5. Affinity constants were obtained using Biacore Insight (Cytiva) using a 1:1 binding kinetics model.

