# OpenReview forum: "Protein Design with Guided Discrete Diffusion"
_NeurIPS.cc/2023/Conference — NeurIPS 2023 spotlight_

### Official Review · Reviewer_EUPz · 2023-07-05

**Soundness:** 3 good
**Presentation:** 3 good
**Contribution:** 3 good
**Rating:** 5
**Confidence:** 2

**Summary:**

This paper is focused on the task of multi-objective protein sequence design using guidance in diffusion models. Since protein sequences are discrete objects, classifier guidance tools from continuous diffusion are not directly applicable. To bridge this gap, the paper proposes a new method NOS, where the classifier guidance is applied to the latent representation of the sequence rather than the discrete sequence. Empirical evaluations on multiple real world sequence design tasks showcase the improved performance of their proposed methods.

**Strengths:**

1. The paper is well written and easy to follow. The technical choices made across the paper are well motivated.

2. The biggest strength of the paper are the empirical evaluations on real world sequence design tasks. In particular, the antibody lead optimization experiment has both an in-silico component and an in-vitro component, with experimental validation of predicted model designs.

**Weaknesses:**

1. The technical contributions of the paper are largely incremental. Upon first reading the paper abstract, it appeared as though the paper proposes a new guidance method to work directly with discrete protein sequences. However, the guidance was applied to the continuous latent representation of the sequence. This is a natural and elegant direction to investigate, but diminishes the technical novelty.

2. Some questions pertaining to the experimental evaluation:
    * In the antibody lead optimization task, could the authors add other baselines such as DiffAb / RFDiffusion?

**Questions:**

Please refer to the weaknesses section above.

Edit: I have read the authors rebuttal. Post discussion with the authors, some aspects associated with novelty and technical contributions have been made clearer. I raised my score by 1 point, but not more as I am not an expert in Bayesian Optimization and some of the contributions of the work are associated with this field.

**Limitations:**

The authors have addresses limitations and proposed directions for future work.

---

> ### Author Rebuttal · Authors · 2023-08-10
>
> Thank you for your review. We have added DiffAb and RFDiffusion comparisons to the antibody lead optimization task in the rebuttal PDF. There we show that our method generates samples with much higher values of the multi-objective acquisition function than these competing diffusion methods. This result mirrors our finding on the percentage of beta sheets and solvent-accessible surface area (SASA) optimization tasks.
>
> **Technical Novelty**
>
> Regarding the technical novelty of our guidance method, we think there might be a small misunderstanding that should be clarified. In both the discrete and continuous diffusion, there is a neural network parameterizing a categorical distribution over tokens, $p(w | h_t)$. We bias the logits of this categorical distribution by taking gradient steps in the hidden units of the network. We are therefore always working with categorical distributions and guiding categorical distributions, though the corruptions are sometimes continuous and sometimes discrete. Performing guidance in this way is distinct from performing guidance directly over a continuous input space and converting this guided variable into discrete tokens. Guiding the categorical distributions directly avoids many of the common pitfalls of converting between continuous and discrete variables, for example the need for rounding continuous vectors to the nearest embedding value, as in [1].
>
> Although we think there is plenty of novelty in our method of guiding categorical distributions, there is also plenty of additional novelty in the many other tools we introduce in our paper, including our method of using saliency maps for selecting edit positions, multi-objective Bayesian optimization as guided generative modeling, and partial deep ensembles for multi-objective surrogate models. All of these insights were necessary to achieve our strong in vitro results, in which we created real proteins with improved binding affinity.
>
> In general we find the notion of novelty implied by the reviewer to be overly narrow in scope. To the best of our knowledge, we are first to perform in vitro experiments with a guided diffusion model operating directly on protein sequences. While discrete diffusion models have not seen major adoption in language modeling, they are poised to have a large impact in protein engineering, and our paper outlines precisely how to accomplish this end, with key details at every level of abstraction, from core machine learning methodology to task specific considerations.
>
> **Additional Experiments**
>
> As the lack of DiffAb and RFDiffusion baselines was one of your major concerns about the submission, we hope you’ll consider raising your score as a response to our extended experiments. As you noted in your review, out empirical evaluations are quite extensive (and have only become more comprehensive with the extended results), and there is a mismatch between this description and the score rubric description, “reasons to reject, e.g., limited evaluation, outweigh reasons to accept”.
>
> [1] Li, Xiang, et al. "Diffusion-lm improves controllable text generation." Advances in Neural Information Processing Systems 35 (2022): 4328-4343.

---

> > ### Comment · Reviewer_EUPz · 2023-08-16
> > **Thank you for your responses - Clarifications provided**
> >
> > First of all, I would like to thank the authors for their extensive efforts during the response period. Follows a point by point response to your rebuttal
> >
> > **Technical Novelty**
> >
> > Yes, I understand that in both the continuous and discrete diffusion settings, there is a neural network parameterizing a categorical distribution over tokens $p(w | h_t)$. The confusion on my end was created by Eq. (4), where $h_t$ was being updated during sampling) instead of $w_t$ being updated during sampling, as was my expectation upon initially reading the paper. The clarification provided in this regard was helpful, and could be added in the main text and abstract to adjust the readers expectations - just a line saying we do guided discrete diffusion by adjusting the categorical distributions would be enough.
> >
> > The comment on limited technical novelty was not intended to come across as harsh, especially since I believe the adopted approach is elegant and has enough empirical support. But I do think that the writing in the initial paragraphs of Section 4.2 can be improved to highlight in a stronger way the contributions concerning "multi-objective Bayesian optimization as guided generative modeling", and "partial deep ensembles for multi-objective surrogate models" and their connection to experimental improvements. As a non-expert in Bayesian optimization, My initial impression upon reading that section was that some of these things are quite standard in the bayesian optimization community, when infact they do not seem to be.
> >
> > **Additional Experiments**
> >
> > Thank you for the additional experiments. The in-vitro experiments were indeed the highlight of the paper, further strengthened by these additional comparisons. I am happy to raise my score to 5, where the score rubric says - Technically solid paper with reasons to accept outweighing reasons to reject.

---

> > > ### Author Response · Authors · 2023-08-18
> > >
> > > Thanks for your thoughtful engagement in the discussion! We believe our paper contains many novel methodological ideas and experimental findings that would be of great interest to the NeurIPS community, and we would be very excited to share them!
> > >
> > > Your feedback regarding the presentation of the novel aspects of our method that are not specific to guided discrete diffusion (e.g. automatic edit position selection via feature attribution, partial deep ensembles for uncertainty quantification from shared representations, etc.) is very helpful, and can be addressed in the camera-ready since we will be allowed an additional page of content in the main text. Indeed, these contributions were originally in the main text but were moved to the appendix in later edits because of the submission page limit.
> > >
> > > Our recent comparison to DiGress (see responses to N1y4) demonstrates that even among guided discrete diffusion methods superficially small differences can lead to a substantial difference in results. Methodological novelty lies not only in high-level ideas but also in design choices and implementation details down to the lowest level of abstraction. Experimental novelty is another dimension entirely. To our knowledge, our submission would be among the first ML conference papers to combine _in silico_ evaluation of genetic algorithms, MCMC algorithms, structure generation, and sequence generation along with further _in vitro_ validation in a real-world setting (design of therapeutic antibodies).
> > >
> > > We greatly value your feedback and believe we have addressed all of the weaknesses enumerated in your review. Would you kindly specify any remaining concerns you have that weaken your support for the paper so we have the opportunity to address them? If you have no major concerns remaining, please consider updating your score to at least 6 to reflect that state.

---

### Official Review · Reviewer_gzgm · 2023-07-06

**Soundness:** 3 good
**Presentation:** 3 good
**Contribution:** 2 fair
**Rating:** 5
**Confidence:** 3

**Summary:**

This paper proposes a discrete diffusion model guided by protein properties. They add a property predictor to the diffusion model, sharing the hidden states. The method is largely inspired by previous work, while experimental results are positive.

**Strengths:**

- The paper is well written.
- The experimental results are convincing and good.
- The function-guided protein design task is important.

**Weaknesses:**

- Many baselines addressing a similar problem are overlooked [1-4].
- The experimental evaluation is limited to a single attribute. The authors could extend the method to multi-objective optimization.



[1] Kirjner, Andrew, et al. "Optimizing protein fitness using Gibbs sampling with Graph-based Smoothing." arXiv preprint arXiv:2307.00494 (2023).

[2] Sinai, Sam, et al. "AdaLead: A simple and robust adaptive greedy search algorithm for sequence design." arXiv preprint arXiv:2010.02141 (2020).

[3] Ren, Zhizhou, et al. "Proximal exploration for model-guided protein sequence design." International Conference on Machine Learning. PMLR, 2022.

[4] Brookes, David, Hahnbeom Park, and Jennifer Listgarten. "Conditioning by adaptive sampling for robust design." International conference on machine learning. PMLR, 2019.

**Questions:**

- Q1: Have you performed in vitro experiments or just evaluated the designed sequence by another neural model?

- Q2: Could you compare your method with more baselines [1-4]?

- Q3: Could you extend the method to multi-objective optimization?

[1] Kirjner, Andrew, et al. "Optimizing protein fitness using Gibbs sampling with Graph-based Smoothing." arXiv preprint arXiv:2307.00494 (2023).

[2] Sinai, Sam, et al. "AdaLead: A simple and robust adaptive greedy search algorithm for sequence design." arXiv preprint arXiv:2010.02141 (2020).

[3] Ren, Zhizhou, et al. "Proximal exploration for model-guided protein sequence design." International Conference on Machine Learning. PMLR, 2022.

[4] Brookes, David, Hahnbeom Park, and Jennifer Listgarten. "Conditioning by adaptive sampling for robust design." International conference on machine learning. PMLR, 2019.

---

> ### Author Rebuttal · Authors · 2023-08-10
>
> Thank you for your review and suggestions for improved comparisons.
>
> ADDITIONAL EXPERIMENTS
>
> We have performed experiments with two of the papers you reference, and the results are quite positive in favor of our method. The detailed results are contained in the rebuttal PDF, but the findings can be summarized as follows. When given the same number of optimization steps, LaMBO-2 can achieve values of the objective function that are uniformly better than PEX, but slightly worse than AdaLead. If we examine the types of solutions these three algorithms find, however, we see the LaMBO-2’s samples are much more natural, and therefore likely to be useful in practice. When calibrated against this consideration, LaMBO-2’s samples achieve much higher values than samples from PEX or AdaLead with comparable naturalness. We also find that LaMBO-2 adds a minor computational overhead for these significant gains in sample quality. We encourage you to examine the detailed explanation in the attached PDF.
>
> IN-VITRO AND MULTI-OBJECTIVE RESULTS
>
> We would like to re-emphasize that our work already includes both in vitro experiments (Figure 7 right) and multi-objective evaluations (Figure 7 left), in which we optimize for both expression yield and binding affinity. Please refer to Section 5.4 and Appendix E5, where we have detailed explanations of the multi-objective evaluation and in vitro experiments. Indeed, our method is specifically targeted at both multi-objective use cases and practical considerations for in vitro protein design, as described in Section 4.2. These considerations are also described at length in Appendix E1, where we provide an introduction to multi-objective Bayesian optimization, and Appendix E4, which includes a detailed description of the training data and methods designed specifically for the in vitro trials.
>
> ADDITIONAL BASELINES AND CONCURRENT WORK
>
> While Kirjner et al also looks like a relevant baseline for this task, it was made public after the NeurIPS submission deadline. For the purposes of this conference, we don’t think it’s reasonable to expect a comparison, and we leave it for future work. We also leave a comparison with Brookes et al. (CbAS) for future work but note that evaluations in the PEX suggest it is a much stronger baseline than CbAS on all tasks considered in the PEX paper.
>
> ADDENDUM: EXPERIMENT DETAILS FOR FIGURE 2
>
> Here are some additional experimental details as promised in the PDF
> For LaMBO-2 we took a sampling step to be one diffusion step, each of which consisted of 16 gradient updates.
> PEX and AdaLead perform local search in sequence space on a learned approximation of the goal objective, akin to genetic optimization with a learned value function.
> For AdaLead a sampling step is one survival and recombination event, followed by 32 x 16 = 512 model evaluations on random mutants of the recombinants.
> Likewise a sampling step for PEX is a contraction of the population to the proximal frontier, followed by 512 evaluations of random mutants along the frontier.
> For both genetic methods we used the authors' recommended hyperparameters.

---

> > ### Comment · Reviewer_gzgm · 2023-08-14
> > **Response to authors**
> >
> > Thanks for the author's response.
> >
> > I've not read the full rebuttal; however, I appreciate the author's effort. I would like to answer:
> >
> > - How long do in vitro experiments take?
> > - Could the author provide an anonymous link to the code base for further checking?
> >
> > By the way, I've raised the score from 4 to 5.

---

> > > ### Author Response · Authors · 2023-08-14
> > >
> > > thanks for your engagement during the discussion period!
> > >
> > > - after we specify the antibodies we wish to validate _in vitro_ it usually takes about 4 weeks to get expression measurements, and an additional two weeks after that to get affinity measurements.
> > >
> > > - the discussion guidelines prohibit sharing the link here, but you can find the link to the anonymized code at the bottom of page 2 of the original submission.
> > >
> > >
> > > We appreciate your willingness to revise your score, and hope you will consider raising it further. Our response has addressed both of the concerns you raised in your original review. In particular we added the comparison to AdaLead and PEX specifically at your request. Could you please clarify any remaining concerns you have that weaken your support for acceptance?

---

### Official Review · Reviewer_N1y4 · 2023-07-07

**Soundness:** 3 good
**Presentation:** 3 good
**Contribution:** 3 good
**Rating:** 6
**Confidence:** 5

**Summary:**

This paper proposes new methods for discrete diffusion guidance, making it possible to optimize protein sequences for local and global properties while retaining high sequence likelihood.

**Strengths:**

1. The paper addresses a problem closely related to the NeurIPS community.
2. The methods sound reasonable.
3. I think the introduction is well-written, especially the second paragraph, which effectively highlights the significance of protein sequence design.
4. The experimental results are also significant.

**Weaknesses:**

1. Overall, I find the presentation of the paper quite comfortable, but I feel that the reading experience is slightly compromised due to some details not being included in the main text.
2. Lack of the comparison with the guidance for discrete diffusion in [1].

**Questions:**

1. I looked at the supplementary materials, and it seems that the selection of edit positions is fixed after the initial determination. What would happen if the positions were reselected at each denoising step?
2. There is another method for discrete guidance [1]. I would like the authors to compare the results of this method with the proposed approach.
3. This seems to resemble an optimization problem where a seed sequence is given, and during inference, only a few positions are edited for amino acid types. I'm curious about the training process. Was the training performed on the entire sequence?
4. The notation $\mathcal{D}$ in Line 168 is not specified or defined.
5. The notation $d$ in Eq.5 is not specified or defined. I guess it may represent the dimension of embeddings.
6. $C_{300}^{8}$ instead of $C_{8}^{300}$ in Line 260?

I would be happy to revise my score if the authors can provide satisfactory responses.

**Minor Concern:**
1. Equation 1 has an extra comma at the end.
2. The notation $L$ in the article is ambiguous, as it is used in two different contexts. In one instance, it represents the length of the sequence, while in another instance, it represents the depth of the encoder network.

**Limitations:**

The paper lacks a discussion of the limitations of the work. I hope the authors will address the limitations and include this discussion in the revised version of the paper during the rebuttal period.

**Reference:**
1. Vignac, Clement, et al. "DiGress: Discrete Denoising diffusion for graph generation." The Eleventh International Conference on Learning Representations. 2022.

---

> ### Author Rebuttal · Authors · 2023-08-10
>
> Thank you for your detailed response and suggestions.
>
> **DiGress Comparison**
>
> While DiGress looks like a closely related and compelling method, its application appears to be limited to graph representations of small molecules, with a maximum of 9 nodes. As our proteins have 1000 nodes, adapting DiGress to our setting would be highly non-trivial and is an effort better to future work, and potentially its own entire paper. On the point of extending baseline comparisons, however, we would like to point out the extended experiments included in our rebuttal PDFs. In those experiments, we compare our method with AdaLead and PEX, which are two popular methods targeted specifically at protein design. We also provide more comparisons with competing diffusion methods (DiffAb and RFDiffusion) that were missing from the earlier protein design experiments.
>
> To answer your itemized questions
> 1. Integrating selection of the edit locations into the diffusion process is a very interesting idea. The challenge with selecting positions at every iteration would be maintaining the constraint that N residues must be conserved (remain unchanged). If new locations could be chosen at every point in the diffusion, the union of all the chosen locations could be much larger than the budget after several iterations. In an unconstrained protein design setting, this effect might be desirable, the design process could iterate over the entire protein throughout the diffusion process, but this freedom is not desirable in many practical applications.
> 2. DiGress does not appear to be directly applicable to proteins, which tend not to be represented as graphs, and which are much larger than small molecules.
> 3. The training process is performed on the entire sequence. Both the generative model and discriminative model are trained to take a corrupted version of the entire sequence as input. During sampling, corrupts are applied along the entire sequence as well, even when infilling only a small region. See Algorithm 1 and Figure 8 in the Appendix for a more detailed walkthrough of the procedure and a visualization. Applying noise along the entire length of the sequence ensures there is no distribution shift between the training procedure and any possible infilling size, but global consistency is not hurt by this procedure, as later steps in diffusion have very small corruptions, and the majority of sequence is passed to the model uncorrupted, informing the new amino acid values that are sampled in the infilling region. As an alternative to training with corruptions on the entire sequence, one can also sample masks, and corrupt only the masked region during training (e.g. [1]). The perk of our method as compared to the limited masking approach is that our models can be used with any masking size or location, as opposed to being trained for a particular masking distribution.
> 4. Thank you for catching this detail. D refers to the dataset and is a commonly used notation in Bayesian inference, where we frequently define posterior distributions over functions conditioned on a given dataset. We will clarify this in a camera-ready.
> 5. You are correct that d is the dimension of the hidden representation $h_0$. This will also be clarified in a camera-ready.
> 6. Both $C^n_k$ and $C^k_n$ are used and there is no universal standard. We will change the “choose” notation to binomial coefficient notation in order to remove any possible ambiguity.
>
> We also thank you for the additional small suggestions listed under “Minor Concerns”.
>
> **Limitations**
>
> As it is not possible to upload a revised draft of the paper, we include a new limitations section below, which will be included in a camera-ready version of the paper:
>
> Limitations: Methodologically, one of the most important limitations of the paper is that it cannot easily leverage pre-training on general protein families before fine-tuning on any particular protein family. This limitation is the result of our use of alignments to create fixed-length sequences. These alignments are applicable to families of proteins with a large number of sequences (hundreds of thousands to millions of sequences), but do extend easily across families. The solution to this limitation would be to extend our method to diffusions that generate variable-length sequences, for example by applying insertions and deletions during the corruption process. Experimentally, the most notable limitation of the current work is the fact that we only evaluate the design of antibodies. Our method is applicable to other common protein families, and other biological sequences, and we hope to demonstrate that generality in future work.
>
> We hope that you will consider raising your score in response to answers and additional experiments. If you have any followup questions about our responses, let us know and we will provide further clarification.
>
> [1] Saharia, Chitwan, et al. "Palette: Image-to-image diffusion models." ACM SIGGRAPH 2022 Conference Proceedings. 2022.

---

> > ### Comment · Reviewer_N1y4 · 2023-08-13
> >
> > Thank you for your response.
> >
> > Overall, I appreciate this paper's content, and I have given positive evaluations for Soundness, Presentation, and Contribution. However, my primary concern still revolves around the lack of a comparison with DiGress.
> >
> > Firstly, there is an issue with the following statement from the authors in the general response:
> > > Two of the baselines suggested by reviewers were ArXived after our submission date and should be considered concurrent work [4,5].
> >
> > The "5" mentioned above is the original DiGress paper I referred to. DiGress was submitted to ICLR 2023 and has already been accepted. Since ICLR's review process is fully open, the paper has been accessible on the OpenReview platform since late 2022, and the results of ICLR have been publicly available for several months. Hence, it cannot be considered concurrent work. The authors' mention of submitting to ArXiv after the NeurIPS 2022 deadline is true only for the fourth version; the previous three versions were submitted earlier and should not be considered concurrent work. Moreover, the differences between the latest version and the previous three are not significant.
> >
> > DiGress introduces a discrete guidance approach for diffusion models, and although their experiments focus on small molecule generation, the presented guidance itself is not inherently limited to small molecules. If you examine the formulas related to the guidance in the paper, you'll notice that the only thing specific to small molecules is the input graph, but the method itself is not constrained.
> >
> > The authors' response does not adequately address my concern. Therefore, I intend to maintain my original score.

---

> > > ### Author Response · Authors · 2023-08-13
> > > **comparison with DiGress**
> > >
> > > Thanks for your timely engagement during the discussion period, and for your clarification on the timeline of the publication of DiGress. Considering the large number of orthogonal contributions our paper provides (e.g comparisons to samples from 3D structure models and genetic methods) it is a little hard to understand making this single comparison the determining factor of your overall score. To be absolutely clear, are you saying that if we were to provide a comparison with DiGress this week you would be open to a substantial increase in your score, e.g. to a 7 (assuming you were satisfied with the quality of the comparison)?

---

> > > > ### Comment · Reviewer_N1y4 · 2023-08-14
> > > >
> > > > Because the central focus of this paper lies in introducing guidance for discrete diffusion, which is also the primary claim of your contribution. DiGress, in this regard, **serves as the most direct and relevant baseline for comparison**. For me, the other supplementary baselines like RFDiffusion and DiffAb are not nearly as crucial as DiGress.
> > > >
> > > > Therefore, this is why I have consistently anticipated the authors to provide comparison results with DiGress. Furthermore, DiGress's methodology is versatile and is not confined to small molecules; it also does not fall under the category of concurrent work. If the authors can supply supplementary experiments that convincingly demonstrate the superior effectiveness of their method compared to DiGress, I would be willing to adjust my score to 6. If considering a score of 7, I will need to contemplate it further.

---

> > > > > ### Author Response · Authors · 2023-08-17
> > > > > **DiGress comparison (part 1)**
> > > > >
> > > > > We have performed a comprehensive comparison between DiGress’s guidance mechanism and NOS. Thank you for pointing out this facet of the DiGress method. It is indeed a related approach and a sensible baseline for our discrete guidance approach. In what follows, we will use DiGress to refer to the guidance method used in Vignac et al. 2023, even though this guidance method could be used with discrete diffusion models that are not designed for graph-structured data. We find that while DiGress is an effective guidance method, leading to improved values of the objective function, NOS is superior along two important dimensions: 1. NOS generates samples with higher objective values than DiGress 2. NOS generates samples with better or comparable likelihoods. In particular, for samples in a fixed range of objective values, the likelihoods of NOS samples tend to be higher than DiGress.
> > > > >
> > > > > DiGress can be used with any diffusion model with one-hot encodings and discrete corruptions, and the sampling algorithm can be described as follows (using the notation from our submission). At each denoising step $t$, we use the one-hot encodings as a continuous variable and construct a perturbation distribution from a learned discriminative model $\hat{v} = v_{\theta}(w_t)$,
> > > > >
> > > > > $p_{\theta}(\hat{v} | w_{t-1}) \propto \exp( \lambda \langle \nabla_{w_t} \hat{v}, w_{t-1} \rangle)$
> > > > >
> > > > > In Vignac et al. 2023, the goal is to make predicted properties $\mathbf{\hat{y}}$ as close as possible to a desired properties $\mathbf{y}$, yielding
> > > > >
> > > > > $p_{\eta}(\mathbf{\hat{y}} | G^{t-1}) \propto \exp(-\lambda \langle \nabla_{G^t} ||\mathbf{\hat{y}} - \mathbf{y}||^2, G^{t-1} \rangle)$
> > > > >
> > > > > In our case, we want to maximize the predicted objective values, $\hat{v}$, and thus the sign inside the exponent is flipped. Having constructed the perturbation distribution, we then sample the next value from the base diffusion transition $p_{\theta}(w_{t-1} | w_t)$ perturbed with $p_{\theta}(\hat{v} | w_{t-1})$,
> > > > >
> > > > > $w_{t-1} \sim p_{\theta}(w_{t-1} | w_t) p_{\theta}(\hat{v} | w_{t-1})$
> > > > >
> > > > > The key details for guided sampling can be found in the DiGress code repo (in the file guidance_diffusion_model_discrete.py on line 556), where we see that the guided distribution is the normalized product of the original denoising distribution and the softmax of the gradients scaled with $-\lambda$.
> > > > >
> > > > > On a theoretical level, this guidance has noticeably different properties from NOS. For large $\lambda$, the perturbation $p_{\theta}(w_t | \hat{v})$ collapses to a one-hot on the token index with the largest gradient value. For small values of $\lambda$, $p_{\theta}(w_t | \hat{v})$ becomes a uniform distribution. Therefore $\lambda$ interpolates $p_{\theta}(w_{t-1} | w_t)$ between the original unguided distribution and a one-hot in the max gradient direction. NOS also reduces to unguided infilling when $\lambda = 0$, but $\lambda >0$ only modulates the direction of the gradient update. The distance between the guided and unguided distribution is controlled by the number of langevin steps and the step size hyperparameter $\eta$. Digress amounts to a single update step applied directly to the output token probabilities using a continuous relaxation of the one-hot encoded input, whereas NOS performs a sequence of local updates to hidden states that are actually continuous.

---

> > > > > > ### Author Response · Authors · 2023-08-17
> > > > > > **DiGress comparison (part 2)**
> > > > > >
> > > > > > To perform our comparison, we trained 3 models on each of the two in silico tasks (% Beta sheets and SASA)
> > > > > > - NOS-C [gaussian corruptions + learned embeddings]
> > > > > > - NOS-D [discrete (mask) corruptions + learned embeddings]
> > > > > > - DiGress [discrete (mask) corruptions + fixed one-hot encodings]
> > > > > >
> > > > > > All models use the same backbone transformer and regression heads, facilitating an apples-to-apples comparison.
> > > > > >
> > > > > > For each model we draw 10 samples for each of the 5 seeds, infilling CDRs H1 & H2 & H3 (as in the submission). For DiGress, we perform sampling for large range of scaling values
> > > > > >
> > > > > > $\lambda \in [1e5, 3e4, 1e4, 3e3, 1e3, 3e2, 1e2, 3e1, 1e1, 1e0, 1e\text{-}1, 1e\text{-}2, 1e\text{-}3 ].$
> > > > > >
> > > > > > For NOS, we use $\eta = 1.0$, $K = 10$, and
> > > > > >
> > > > > > $\lambda \in [1e1, 1e0, 1e\text{-}1, 1e\text{-}2, 1e\text{-}3, 0 ].$
> > > > > >
> > > > > > For each model, $\lambda$ modulates the degree to which the model prefers greedy sampling from the value function gradient.
> > > > > >
> > > > > > We evaluate the samples with four metrics:
> > > > > > - (OV-Max) Max value of the ground truth objective function
> > > > > > - (LL-Max) Log likelihood of the sample with the max objective function value
> > > > > > - (OV-HD) Average objective value of samples in the highest decile (0.9 - 1.0) of the objective function distribution, as given by samples from all methods.
> > > > > > - (LL-HD) Average log likelihood of samples in the highest decile, as above
> > > > > >
> > > > > > **% Beta Sheets Task**
> > > > > >
> > > > > > | Model | OV-Max | LL-Max | OV-HD | LL-HD |
> > > > > > |:-------------|---------:|---------:|---------:|---------:|
> > > > > > | DiGress | 0.300429 | -559.062 | 0.251327 | -521.476 |
> > > > > > | NOS-C | 0.356223 | -534.765 | 0.266072 | -485.307 |
> > > > > > | NOS-D | 0.360515 | -530.673 | 0.264327 | -478.523 |
> > > > > >
> > > > > > **SASA Task**
> > > > > >
> > > > > > | Model | OV-Max | LL-Max | OV-HD | LL-HD |
> > > > > > |:-------------|---------:|---------:|--------:|---------:|
> > > > > > | DiGress | 12281.1 | -567.388 | 12079.1 | -533.607 |
> > > > > > | NOS-C | 12533.7 | -498.372 | 12149.5 | -518.321 |
> > > > > > | NOS-D | 12542.1 | -549.382 | 12131.5 | -545.353 |
> > > > > >
> > > > > > From these numbers, we can see that NOS generates samples with higher values of the objective while also keeping likelihood values high. As tables only capture a subset of the information in the original samples, we will include density plots in the camera-ready version of the paper.

---

> > > > > > > ### Comment · Reviewer_N1y4 · 2023-08-18
> > > > > > >
> > > > > > > Thank you for your response! I'm pleased to see that you promptly provided comparisons with DiGress. I appreciate that you explained the experimental design and the selection of hyperparameters in detail. The additional experimental results sufficiently demonstrate the superiority of your method. Since my main concern have been addressed, I've decided to raise my score to 6.

---

> > > > > > > > ### Author Response · Authors · 2023-08-21
> > > > > > > >
> > > > > > > > We appreciate your support! Thanks for your valuable contributions to this discussion.

---

### Official Review · Reviewer_j6TU · 2023-07-07

**Soundness:** 4 excellent
**Presentation:** 3 good
**Contribution:** 2 fair
**Rating:** 7
**Confidence:** 3

**Summary:**

The submission introduces a new diffusion for protein design. It focuses on overcoming the challenge of discrete nature of protein sequences. The model succeeded in local and global properties optimization of antibody sequences making them more applicable for therapeutic purposes.

**Strengths:**

Originality: Protein diffusion models aren't new (there are many related contribution into this field), however, the discrete nature of protein sequences challenges the usage of these models directly in sequence space, Therefore, the submission is important, novel, and useful. The claims are well supported.
Quality: Overall quality of the paper is high, but it's better to add and extend the comparison to and discussion of other protein diffusion models.
Clarity: The paper is clearly written and contains all the necessary citations. Methods, experiments, and equations are clearly described.
Significance: Diffusion is a promising approach in protein design. The submission provides a valuable impact to optimize protein sequences for local and global properties.

**Weaknesses:**

The submission lack of comparison to currently available deep learning protein design approaches such as RFdiffusion.

**Questions:**

1. Please provide the comparison to RFdiffusion and other available deep learning protein design tools.

**Limitations:**

The paper doesn't describe limitations and future work.

---

> ### Author Rebuttal · Authors · 2023-08-10
>
> Thank you for your supportive review and for emphasizing the impact and novelty of our submission! We have included the requested comparison with RFDiffusion in our rebuttal PDF. We also include DiffAb and several other local sequence search methods (AdaLead, PEX) for the protein design task in our extended experiments. The experiments show that our method is dominant for practical applications of generative design, and we hope these results inspire your continued support for our submission.

---

> > ### Comment · Reviewer_j6TU · 2023-08-14
> >
> > Thank you for the response! I'm satisfied with the changes and additions.

---

### Official Review · Reviewer_UJds · 2023-07-26

**Soundness:** 2 fair
**Presentation:** 1 poor
**Contribution:** 3 good
**Rating:** 6
**Confidence:** 3

**Summary:**

The paper presents a novel approach to applying diffusion models to optimize protein sequences, specifically for antibody design. Diffusion models have been successful in handling continuous data like images, but their direct application to protein structures faces challenges due to limited high-quality structural data. The authors propose a new method for discrete diffusion guidance, enabling the optimization of protein sequences for desired properties while maintaining high sequence likelihood. They further apply this method to a real-world protein design task, focusing on optimizing antibodies for increased expression yield and binding affinity to a therapeutic target while adhering to locality and liability constraints.

**Strengths:**

Novel Ideas and Comprehensive Solution: The method proposed in the paper introduces several novel ideas that address a critical challenge in protein design.

A notable strength of the method is its ability to optimize protein sequences without relying on structural information.Protein design approaches often face limitations due to the scarcity of high-quality structural data and the need to convert structures into sequences for synthesis. By bypassing the requirement for structural information, the proposed method reduces complexity and provides a more straightforward and practical solution for designing proteins.

The absence of a structural component in the optimization process simplifies the method's implementation and reduces computational complexity.

The authors demonstrate the efficacy of their approach through real-world evaluations. By applying the method to optimize antibodies for increased expression yield and binding affinity to a therapeutic target, the paper offers tangible and meaningful results that resonate with practical protein design needs.

One of the noteworthy strengths of the method is its ability to optimize protein sequences while maintaining high sequence likelihood and by extension expression efficiency.

**Weaknesses:**

Unclear Introduction and Insufficient Motivation: The paper's introduction lacks clarity and fails to provide a meaningful digest of key topics in the field of protein design. As a result, the motivation for the research is poorly established, which hampers the reader's understanding and engagement throughout the rest of the paper, making it challenging to appreciate the significance of the presented work.

Lack of Reproducibility: The absence of all test data and failure to provide the code used in the experiments significantly impairs the reproducibility of the results. Reproducibility is a fundamental aspect of scientific research, and the unavailability of essential resources restricts other researchers' ability to validate and build upon the findings.

Missing Computational Cost and Time Information: The paper lacks essential details concerning the computational cost and time required for both training and inference of the proposed method. Such information is crucial for assessing the feasibility and practicality of the approach, and its absence leaves readers with an incomplete understanding of the method's resource requirements.

Inadequate Description of Model Architecture and Training Procedure: Section 4.2 requires more comprehensive information on the model architecture, feature extraction layers, and the encoder used in the proposed method. Similarly, the training procedure and hyperparameter tuning should be thoroughly explained to ensure clarity and reproducibility.

Insufficient Information on Test Data: The paper does not provide sufficient details about the test data used, such as the number of samples and the criteria for their selection. Without this information, it remains unclear if the reported results generalize well to diverse protein sequences.

Lack of Relevant Information in Figure Captions: Figure captions throughout the paper lack relevant information that could aid in the readers' interpretation of the presented results. Clear and informative captions are essential for facilitating a comprehensive understanding of the visual data.

Missing Quality Assessment of IgFold Structures: In section 5.1, the authors fail to report on the quality of the IgFold structures used in structure-based methods. Since the performance of these methods can be highly dependent on the starting structure's quality, this information is crucial for the paper's credibility.

Inadequate Explanation of Comparison Choice: In section 5.3, the rationale behind comparing salient selection with uniform random selection from the entire sequence remains unclear. The authors should provide a more insightful justification for this choice, considering the low probability of randomly selecting critical regions like CDRs.

Lack of Ablation Analysis for Objective Functions: The authors do not thoroughly evaluate the importance of the chosen objective functions through ablations. Specifically, in section 5.2, the effect of 40% beta on ProtG3T Log Likelihood should be explained and analyzed in more detail.

Unexplained Statement on Sensitive Data: The statement regarding non-disclosure of specific drug targets in the experiments lacks a clear explanation. The authors should clarify the reasons behind this decision and provide reassurance regarding the privacy and ethical handling of sensitive data.

Avoidance of Emotional Statements: The use of emotional statements, such as "eager to" or "encouraging to see," should be avoided in scientific writing. Such statements can introduce bias and subjectivity, undermining the paper's objectivity and professionalism.

Minor issues
* Line 46: “there is no analogous method for proteins, which do not encode the same general-purpose semantics. As a result many […]” It is not clear what the authors are saying here. Explain how the encoded semantics in protein space differs from NLP and why other sampling methods have been used to address this issue.
* Figure 1 caption: “[...] a powerful method for protein design.” Please stick to informative text about the figure in the caption.
* Line 84: “A protein is only useful if it can be expressed in living cells [...]” This is incorrect. A protein could be expressed for instance in a cell-free translation system and still be useful.
* Figure 7 caption: Last sentence doesn’t belong in the caption.
* Equation (1), remove Z and replace equation with proportionality or explain what Z is.
* The OAS dataset needs to be briefly described somewhere in the text.


**Questions:**

1. Could you provide a more concise and clear introduction that offers a comprehensive overview of the key topics in the field of protein design? Additionally, could you elaborate on the motivation for the research to better establish the significance of your work throughout the paper?
2. Can you provide detailed information on the computational cost and time required for both training and inference of your proposed method? This will help readers assess the feasibility and practicality of implementing your approach.
3. In section 4.2, could you offer a more thorough explanation of the model architecture, feature extraction layers, and the encoder used in your proposed method? Similarly, can you provide more information about the training procedure and hyperparameter tuning to ensure clarity and reproducibility?
4. Please provide additional details about the test data used, including the number of samples and the criteria used for their selection. This information will help readers understand the generalizability of the reported results to diverse protein sequences.
5. Could you include more relevant information in the figure captions to aid readers in interpreting the presented visual data?
6. Can you report on the quality assessment of the IgFold structures used in structure-based methods in section 5.1? Understanding the starting structure's quality is crucial for evaluating the performance of these methods.
7. Could you clarify the statement regarding the non-disclosure of specific drug targets in the experiments? Please provide an explanation for this decision and assure readers of the ethical handling and privacy of sensitive data.
8. Line 37, what does this mean: “Searching directly in sequence space eliminates the need to recover sequence from structure.”
9. Line 38, any examples (references) for: “ Protein sequence models are also comparatively fast, especially during inference, and can leverage sequence datasets that are often several orders of magnitude larger than their structural equivalents.”


**Limitations:**

The presentation lacks clarity and fails on several points (see weaknesses).

The unavailability of all test data and the absence of provided code significantly impair reproducibility, limiting the validation and extension of the research by other researchers.

The paper does not present detailed information on several topics (see weaknesses).

Ablation studies are lacking (see weaknesses).

---

> ### Author Rebuttal · Authors · 2023-08-10
>
> Thank you for the detailed feedback.
>
> First of all, we would like to push back on the assertion that our paper has inadequate description of models or training procedures or lack of reproducibility. We encourage the reviewer to take a look at the paper appendices and follow the anonymous repo link (https://anonymous.4open.science/r/protein-seq-diffusion-0431/README.md) included on page 2 of the submission.
>
> Answers to questions:
> 1. Currently protein design lacks methods that satisfy the following criteria simultaneously (a) reliable optimization of a goal objective function (b) generated sequences that are likely under the distribution of real proteins (naturalness) (c) fast sampling. Notably, hill-climbing methods often struggle to produce natural sequences, while popular sequence-based methods like MCMC can be slow to converge and computationally expensive. This gap in functionality is the fundamental motivation for our work. While we find this writing feedback helpful, and we have added additional descriptions of our motivation, as you can see in the top-level rebuttal comment, we also think a conference paper can never provide a truly comprehensive introduction to a subject, and this is a task better left to review paper. For protein design, there are many great existing review papers [2,3,4], which discuss the tradeoffs between sequence-based and structural methods and provide extensive context on the challenges of creating design methods that sample from a desired conditional distribution while maintaining speed and trading off exploration-exploitation, as our method is designed to.
> 2. The training process for our method converges in about 100 epochs, which takes about 7 hours on one RTX8000 GPU. Sampling from this model takes a few minutes or less on one GPU for both corruption processes. This level of compute is accessible to most academic and industry labs, and therefore there’s no fundamental barrier to others using or building on our method.
> 3. Please refer to Appendix E.3 in the original submission, which contains all the requested details about model architecture, as well all necessary details of training and sampling, including an example algorithm and the hyperparameters used.
> 4. The precise test seeds were released in the submission, and can be viewed at the following link (https://anonymous.4open.science/r/protein-seq-diffusion-0431/poas_seeds.txt). pOAS is a dataset of around 120,000 paired antibody sequences. For a more detailed description you can also see the following link: https://opig.stats.ox.ac.uk/webapps/oas/documentation_paired. Our test seeds were chosen randomly from pOAS and were removed from the training and validation sets.
> 5. We appreciate your detailed writing feedback, but we see stated criticisms as fairly subjective, and not related to a fundamental lack of clarity or informativeness in the figure captions. If there is missing information that a reader would need for a full understanding of the presented number (for example the meaning of error bars), please point those out in your response, and we will be sure to address them so that there is no room for misinterpretation of the results.
> 6. IgFold reports predicted RMSD (pRMSD) values for its folded structures. For our 10 seed pairs of sequence and predicted structure, the overall average per-residue pRMSD was 0.34. We also include the average and maximum pRMSD values for each individual seed in the table below. The framework typically has a very low pRMSD while the maximum value is located in CDR H3. These values are around the average or well below average compared to values in the IgFold paper [1], and therefore we don’t have any concerns about using these predicted structures as starting points.
> | Seed | 1 | 2 | 3 | 4 | 5 | 6 | 7 | 8 | 9 | 10 |
> | ----------- | ----------- | ----------- | ----------- | ----------- | ----------- | ----------- | ----------- | ----------- | ----------- | ----------- |
> | Avg | 0.3 | 0.43 | 0.27 | 0.46 | 0.27 | 0.29 | 0.29 | 0.29 | 0.26 | 0.56 |
> | Max | 2.05 | 2.95 | 2.27 | 4.32 | 2.30 | 1.67 | 1.68 | 2.37 | 1.32 | 3.99 |
>
> 7. The non-disclosure is the result of intellectual property concerns from our industry collaborator, not ethical or privacy concerns.
> 8. By “searching directly in sequence space” we mean that we never have to produce a structure in the process of designing a protein sequence. The only time we use structure representations is when we fold sequences into structures as part of the evaluation step in our optimization experiments.
> 9. One example is SabDab (a database of structures and sequences) versus pOAS (a database of sequences alone). SabDab has about 6,000 paired antibodies, while pOAS has about 120,000 paired sequences. If we expand the comparison to unpaired structures and sequences it is about 7,500 structures vs 3,549,291,485 sequences. Beyond antibodies, another example is the size of the PBD, which contains about 200,000 structure entries, versus the size of UniProt, which contains 248,000,000 entries.
>
> As your concerns centered primarily on clarity, rather than methodological or experimental soundness of the paper, we hope that you will consider revising your score in response to our detailed answers.
>
> [1] Ruffolo, Jeffrey A., and Jeffrey J. Gray. "Fast, accurate antibody structure prediction from deep learning on massive set of natural antibodies." Biophysical Journal 121.3 (2022): 155a-156a.
> [2] Hummer, Alissa M., Brennan Abanades, and Charlotte M. Deane. "Advances in computational structure-based antibody design." Current Opinion in Structural Biology 74 (2022): 102379.
> [3] Tian, Xiaochen, et al. "SEQUENCE VS. STRUCTURE: DELVING DEEP INTO DATA DRIVEN PROTEIN FUNCTION PREDICTION." bioRxiv (2023): 2023-04
> [4] Hie, Brian L., and Kevin K. Yang. "Adaptive machine learning for protein engineering." Current opinion in structural biology 72 (2022): 145-152.

---

> > ### Comment · Reviewer_UJds · 2023-08-16
> >
> > Thank you for the rebuttal.
> >
> > The authors have addressed various concerns, and I have consequently made adjustments to the rating.

---

> > > ### Author Response · Authors · 2023-08-21
> > >
> > > Thank you for your detailed review, it's important to us to make our work as clear and accessible as possible and your comments have been very helpful. We greatly appreciate your support!

---

### Author Rebuttal · Authors · 2023-08-10

We are glad to share many exciting new results inspired by reviewer comments in the attached PDF, including a new comparison with RFDiffusion and DiffAb on antibody lead optimization (Figure 1) and two additional genetic algorithm baselines mentioned by gzgm (Figure 2). In both of these experiments, we show that our proposed methods (NOS/LaMBO-2) generate better samples than competing methods by optimizing a goal objective without falling off the manifold of plausible sequences.

Our new results further solidify what is already a significant and timely contribution to the field of ML-guided protein design. Two of the baselines suggested by reviewers were ArXived after our submission date and should be considered concurrent work [4,5]. Structural diffusion methods have become a popular approach to generative protein design [1] but come with several drawbacks. Our novel method for guided sequence diffusion provides an alternative to structural diffusion that conserves many of its benefits (e.g. the potential for gradient-guided search) while avoiding several downsides (e.g.  limited structural data or the use of inverse folding models). Our method is also a compelling alternative to sequence hill-climbing [2], which can rapidly fall off-manifold, or MCMC methods [3], which can suffer from slow convergence and high computational overhead. Lastly, it is also noteworthy that among the many recent methods for ML-assisted protein design, we are one of only a handful that provide both open-source in silico evaluations and in vitro experiments.

We would like to note that the majority of reviewers thought our methods were sound and sufficiently novel but wanted to see additional baselines or wanted clarifications on the paper’s text. We have directly addressed many of these concerns through our rebuttal answers and our additional experiments, and we hope that reviewers will be open to revisiting their scores in light of these updates.

REFERENCES

[1] Watson, Joseph L., et al. "Broadly applicable and accurate protein design by integrating structure prediction networks and diffusion generative models." BioRxiv (2022): 2022-12.

[2] Sinai, Sam, et al. "AdaLead: A simple and robust adaptive greedy search algorithm for sequence design." arXiv preprint arXiv:2010.02141 (2020).

[3] Hie, Brian L., et al. "Efficient evolution of human antibodies from general protein language models." Nature Biotechnology (2023).

[4] Kirjner, Andrew, et al. "Optimizing protein fitness using Gibbs sampling with Graph-based Smoothing." arXiv preprint arXiv:2307.00494 (2023).

[5] Vignac, Clement, et al. "DiGress: Discrete Denoising diffusion for graph generation." The Eleventh International Conference on Learning Representations. 2022.

---

### Decision · Program_Chairs · 2023-09-21

**Decision:**

Accept (spotlight)

**Comment:**

The paper makes significant methodological and empirical contributions. The proposed approach of controllable discrete diffusion allows for a principled use of sequence data as structural data is scarce. The AC and reviewers feel that all key concerns have been satisfactorily addressed and this will strengthen the submission. We therefore urge the authors to incorporate the clarifying points and additional experiments (s.a. comparisons with DiGress) made during the discussion phase.